# Spatial signals link exit from mitosis to spindle position

**Jill Elaine Falk[1], Dai Tsuchiya[2], Jolien Verdaasdonk[3], Soni Lacefield[2], Kerry Bloom[3], Angelika Amon[1]\***

[1]David H Koch Institute for Integrative Cancer Research, Howard Hughes Medical Institute, Massachusetts Institute of Technology, Cambridge, United States; [2]Department of Biology, Indiana University, Bloomington, United States; [3]Department of Biology, The University of North Carolina at Chapel Hill, Chapel Hill, United States

**Abstract** In budding yeast, if the spindle becomes mispositioned, cells prevent exit from mitosis by inhibiting the mitotic exit network (MEN). The MEN is a signaling cascade that localizes to spindle pole bodies (SPBs) and activates the phosphatase Cdc14. There are two competing models that explain MEN regulation by spindle position. In the 'zone model', exit from mitosis occurs when a MEN-bearing SPB enters the bud. The 'cMT-bud neck model' posits that cytoplasmic microtubule (cMT)-bud neck interactions prevent MEN activity. Here we find that 1) eliminating cMT– bud neck interactions does not trigger exit from mitosis and 2) loss of these interactions does not precede Cdc14 activation. Furthermore, using binucleate cells, we show that exit from mitosis occurs when one SPB enters the bud despite the presence of a mispositioned spindle. We conclude that exit from mitosis is triggered by a correctly positioned spindle rather than inhibited by improper spindle position.

**\*For correspondence:** angelika@mit.edu

**Competing interests:** The authors declare that no competing interests exist.

## Introduction

Asymmetric cell division is a common characteristic of development and is seen in diverse cell types ranging from *Drosophila* neuroblasts to mammalian oocytes. In order to produce viable progeny with distinct cell fates, asymmetrically dividing cells must coordinate nuclear position with the site of cytokinesis. When the spindle is mispositioned with respect to the cleavage plane, cell cycle progression is delayed until the spindle is correctly aligned. In the budding yeast *Saccharomyces cerevisiae*, coupling exit from mitosis with spindle position is particularly important because the site of cytokinesis forms independently of mitotic spindle position (*Pruyne et al., 2004*). Thus budding yeast has evolved mechanisms that align the spindle and that ensure that cytokinesis only occurs after the nucleus has been partitioned between the mother cell and bud, the future daughter cell (*Bardin et al., 2000*; *Miller et al., 1999*; *Pereira et al., 2000*; *Yeh et al., 1995*).

Budding yeast employs two redundant mechanisms to position the spindle along the mother – bud axis. The first positioning mechanism relies on the type V myosin motor Myo2. Myo2, together with its adaptor Kar9 and the plus-end microtubule binding protein Bim1, positions the pre-ana-phase spindle at the bud neck by pulling cytoplasmic microtubules (cMTs) along actin cables (*Beach et al., 2000*; *Kopecká and Gabriel, 1998*; *Miller et al., 1999*; *Miller and Rose, 1998*; *Palmer et al., 1992*). The second pathway is active during anaphase and requires the minus-end microtubule motor protein dynein, together with its associated coactivating complex dynactin. Dynein, when anchored to the cell cortex by Num1, aligns the spindle by essentially towing it along the cortex of the cell (*Farkasovsky and Küntzel, 2001*; *Heil-Chapdelaine et al., 2000*; *Muhua et al., 1994*; *Yeh et al., 1995*). The consequences of eliminating either positioning pathway

**eLife digest** Most cells duplicate their genetic material and then separate the two copies before they divide. This is true for budding yeast cells, which divide in an unusual manner. New daughter cells grow as a bud on the side of a larger mother cell and are eventually pinched off. To make healthy daughter cells, yeast must share their chromosomes between the mother cell and the bud. This involves threading the chromosomes through a small opening called the bud neck, which connects the mother cell and the bud.

A surveillance mechanism in budding yeast monitors the placement of the molecular machine (called the spindle) that separates the chromosomes before a cell divides. This mechanism stops the cell from dividing if the spindle is not positioned correctly. Two models could explain how an incorrectly positioned spindle prevents budding yeast from dividing. The first model proposes that yeast cells do not divide if protein filaments (called microtubules) touch the bud neck. This only occurs if the spindle is not properly threaded into the bud through the small opening of the bud neck. The second model proposes that specific proteins required for cell division (which are found at the ends of the spindle) are inhibited while they are inside the mother cell. This means that the cell cannot divide until one end of its spindle moves out of the mother cell and into the bud.

Now, Falk et al. report the results of experiments that aimed to distinguish between these two models. First, a laser was used to cut the spindle filaments in live yeast cells. This stopped the filaments from touching the neck between the mother cell and the bud, but did not cause the cell to divide. Therefore, these results refute the first model. Next, Falk et al. generated yeast cells that had essentially been tricked into forming two separate spindles before they started to divide. As would be predicted by the second model, these cells could divide as long as an end from at least one of the spindles entered the bud.

These findings strongly suggest that the second model provides the best explanation for how yeast cells sense spindle position to control cell division. The findings also lend further support to previous work that showed that activators of cell division are found in the bud, while inhibitors of cell division are found in the mother cell. Finally, in a related study, Gryaznova, Caydasi et al. identify a protein at the ends of the spindle that acts like a regulatory hub to coordinate cell division with spindle position. Their findings also suggest that the surveillance mechanism is switched off in the bud to allow the cell to divide.

are minor but cells lacking both pathways are inviable (*Miller and Rose, 1998*). Cells that fail to segregate the nucleus into the bud will arrest in late anaphase with the spindle mispositioned in the mother cell compartment. If the cell manages to correct this positioning defect and threads the nucleus through the bud neck into the bud, it will then disassemble the anaphase spindle and exit from mitosis (*Yeh et al., 1995*).

Yeast cells link spindle position and exit from mitosis through the regulation of the essential phosphatase Cdc14 (*Bardin et al., 2000*; *D'Aquino et al., 2005*; *Pereira et al., 2000*; *Pereira and Schiebel, 2005*). Cdc14 functions to reverse mitotic cyclin-CDK (cyclin dependent kinase) activity by dephosphorylating cyclin-CDK targets as well as by targeting cyclins for degradation (*Jaspersen et al., 1998*; *Visintin et al., 1998*; *Zachariae et al., 1998*). These Cdc14 functions cause exit from mitosis, the final cell cycle transition that encompasses disassembly of the mitotic spindle and cytokinesis (*Stegmeier and Amon, 2004*). Cdc14's essential role in exit from mitosis requires that its activity is tightly regulated. Cdc14 is kept inactive from G1 to metaphase by its inhibitor Cfi1/Net1, which functions by sequestering the phosphatase in the nucleolus. It is only upon anaphase entry that Cdc14 is released from Cfi1/Net1 to spread throughout the cell where it antagonizes mitotic CDK activity and so returns the cell to G1 (*Shou et al., 1999*; *Visintin et al., 1999*).

Two pathways control the activity of Cdc14: the Cdc14 early anaphase release network (FEAR) and the mitotic exit network (MEN). The FEAR network serves to ensure anaphase spindle stability, spindle midzone assembly and proper rDNA segregation by transiently releasing Cdc14 from its inhibitor in the nucleolus during early anaphase (reviewed in *Rock and Amon [2009]*). While not essential, this brief release of Cdc14 serves to ensure proper anaphase timing and primes the cell to

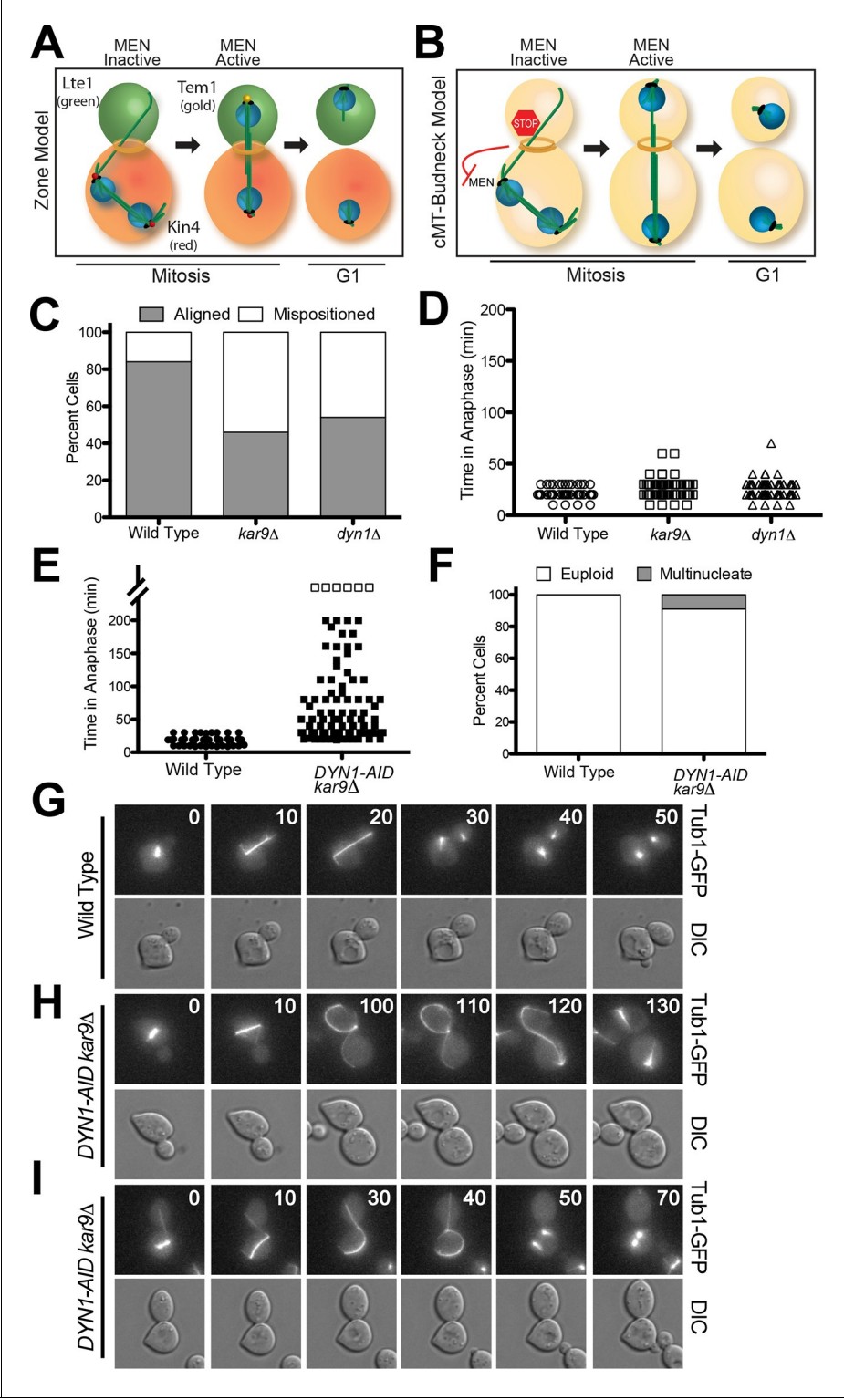

**Figure 1.** A system to induce spindle misposition. (**A**) Zone model of exit from mitosis. Yeast cells are partitioned into two zones: a MEN inhibitory zone in the mother cell compartment (red) and a MEN activating zone in the bud cell compartment (green). If the spindle becomes misaligned in the inhibitory zone, MEN inhibitors such as Kin4 prevent Tem1 enrichment on SPBs thereby inhibiting exit from mitosis. It is only once one SPB escapes the MEN inhibitory zone and moves into the bud cell compartment that Tem1 can become enriched at the daughter-bound SPB and the cell can exit mitosis. Note that in this model, Tem1 is shown not to localize to SPBs in cells with mispositioned spindles. This is based on the observation that Tem1-13MYC does not localize to SPBs in cells with mispositioned spindles (*D'Aquino et al., 2005*). (**B**) cMT - budneck model of exit from mitosis. If the

*Figure 1 continued on next page*

*Figure 1 continued*

spindle becomes misaligned in the mother compartment, cytoplasmic microtubules activate a checkpoint response through their interactions with factors at the bud neck. Once the spindle has realigned, the cytoplasmic microtubules are no longer in contact with the bud neck, the checkpoint signal is eliminated and cells exit from mitosis.(C–D) Wild type (A33138), *kar9Δ* (A33729) and *dyn1Δ* (A32922) cells harboring GFP-tagged α-tubulin were grown to mid-log in YEPD and arrested in G1 with 10 µg/mL of the α-factor pheromone at 25°C. The cultures were released into the cell cycle in YEPD and then loaded onto a Y04C CellASIC flow cell. Cells were imaged on the flow cell in synthetic complete pH 6.0 medium. (C) Quantification of the percent of anaphase cells which misposition their anaphase spindle. Anaphase was defined as any spindle measuring >2 µm. Aligned spindles were defined as those that entered anaphase with one spindle pole in the bud cell compartment. Mispositioned spindles were defined as those that entered anaphase with both spindle poles the mother cell compartment. (D) Time-lapse analysis of anaphase length. n =100 cells for each strain (E–I) *osTIR1* (A35699) and *osTIR1 DYN-AID kar9Δ* (A35707) cells expressing GFP-tagged α-tubulin were grown in YEPD medium at 25°C and arrested in the G1 phase of the cell cycle with 10 µg/mL α-factor pheromone. Cells were released into the cell cycle in YEPD pH 6.0 medium and then monitored by live cell microscopy. Depletion of *dyn1-AID* was induced on a Cellasic flow cell with 100 µM auxin in SC pH 6.0 medium at 25°C. (E) Time-lapse analysis of anaphase length. Open squares indicate cells arrested in anaphase for more than 200 min. (F) Analysis of ploidy. Cells that were arrested and contained a misaligned spindle or cells that exited mitosis that contained an aligned spindle were categorized as 'euploid'. Cells that inappropriately exited mitosis and broke down the spindle in the mother cell compartment were categorized as 'multinucleate'. n=100 cells. (G–I) Montage of representative time-lapse images. The numbers at the top of the GFP images are time in minutes.

efficiently exit from mitosis. In contrast, the MEN is responsible for sustained Cdc14 release during later stages of anaphase and is essential for cells to exit from mitosis (*Jaspersen et al., 1998*; *Lee et al., 2001*; *Stegmeier et al., 2002*; *Visintin et al., 1998*; *1999*).

The MEN is a GTPase signaling pathway whose constituents primarily localize to spindle pole bodies (SPBs; yeast centrosomes). Regulation of the GTPase Tem1 is central to MEN control. When Tem1 is in its GTP-bound state, the MEN is active and cells will exit from mitosis (*Scarfone and Piatti, 2015*). Likewise when Tem1 is inactive, the MEN is off and cells will arrest in anaphase (*Geymonat et al., 2002*; *Shirayama et al., 1994*). Tem1 regulates a kinase cascade comprised of the PAK-like kinase Cdc15 and the protein kinase Dbf2. Tem1 activates Cdc15 by recruiting it to the spindle pole body (*Rock and Amon, 2011*; *Visintin and Amon, 2001*). Cdc15 in turn recruits Dbf2 to spindle poles by creating a phospho-peptide binding domain on the SPB component and MEN scaffold Nud1. In its phosphorylated state, Nud1 docks the Dbf2-activating subunit Mob1 (*Rock et al., 2013*). Activated Dbf2-Mob1 together with Cdc5 then promote the sustained release of Cdc14 from the nucleolus through a largely uncharacterized mechanism (*Manzoni et al., 2010*; *Mohl et al., 2009*).

Tem1 itself is controlled by two opposing factors, Bub2/Bfa1 and Lte1. Bub2/Bfa1 functions as a GTPase-activating protein complex (GAP) for Tem1 and so inhibits the MEN (*Bloecher et al., 2000*; *Geymonat et al., 2002*; *Li, 1999*; *Shirayama et al., 1994*). The GAP complex in turn is regulated by the protein kinase Kin4. Kin4 localizes to the mother cell cortex as well as the mother cell-localized SPBs and functions to maintain GAP activity by preventing the inactivation of Bub2/Bfa1 by the Polo kinase Cdc5 (*D'Aquino et al., 2005*; *Maekawa et al., 2007*; *Pereira and Schiebel, 2005*). Lte1 localizes to the bud cell compartment and promotes exit from mitosis by preventing Kin4 localization to SPBs in the bud (*Bertazzi et al., 2011*; *Falk et al., 2011*). Lte1 displays homology with guanine nucleotide exchange factors (GEFs); however, whether Lte1 also functions as a GEF for Tem1 remains unknown.

Spindle position regulates MEN activity. When the spindle is mispositioned, the MEN is inactive: Cdc14 is sequestered in the nucleolus and cells arrest in anaphase (*Bardin et al., 2000*). This regulatory mechanism that prevents exit from mitosis in response to spindle misposition is called the spindle position checkpoint (SPoC; [*Muhua et al., 1998*]). Two models have been proposed to explain how spindle position regulates MEN activity. The 'zone model' proposes that the cell is partitioned into a MEN inhibitory zone in the mother cell compartment and a MEN activating zone in the bud (*Figure 1A*) (*Chan and Amon, 2010*). The MEN inhibitor Kin4 localizes to the mother cell, the MEN activator Lte1 to the bud (*Bardin et al., 2000*; *D'Aquino et al., 2005*; *Pereira et al., 2000*; *Pereira and Schiebel, 2005*). In the event that anaphase spindle elongation occurs only in the mother cell, the spindle poles (where Tem1 resides) cannot escape the negative regulation of Kin4 and the MEN is kept inactive. Inhibition of Tem1 is only relieved once the spindle realigns along the mother-bud axis and one spindle pole exits the Kin4 inhibitory zone. The bud compartment promotes Tem1 activation through redundant mechanisms: 1) the bud is largely devoid of Kin4 and 2)

the Kin4 inhibitor Lte1 prevents any small amount of Kin4 present in the bud from localizing to the daughter SPB.

Support for the zone model comes from studies in which the localization of Kin4 and Lte1 have been switched. Targeting Lte1 to the mother cell leads to inappropriate mitotic exit in cells with misaligned spindles (*Bardin et al., 2000*; *Bertazzi et al., 2011*; *Castillon et al., 2003*; *Geymonat et al., 2009*). Targeting Kin4 to the bud and simultaneously inactivating its inhibitor, Lte1, causes anaphase arrest even in cells with correctly positioned spindles (*Chan and Amon, 2010*; *Falk et al., 2011*).

A second model proposes that MEN activity is controlled by a microtubule-based checkpoint mechanism (*Figure 1B*; henceforth the 'cMT - budneck model') (*Adames et al., 2001*; *Moore et al., 2009*; *Muhua et al., 1998*). The model posits that stable contact between cytoplasmic microtubules and the bud-neck activates a checkpoint response that prevents cells from exiting mitosis. The hypothetical cMT checkpoint sensor would, according to this model, localize to the mother side of the septin ring (*Castillon et al., 2003*). The model was proposed based on studies showing that cytoplasmic microtubule loss from the bud neck precedes anaphase spindle disassembly and exit from mitosis (*Adames et al., 2001*; *Moore et al., 2009*). Laser ablation of cytoplasmic microtubules interacting with the bud neck was further reported to trigger exit from mitosis (*Moore et al., 2009*).

Here we describe several experimental approaches aimed at distinguishing between the zone model and the cMT - budneck model. These analyses refute the cMT - budneck model and support the zone model. In the first approach we conducted live cell imaging to investigate the relationship between the presence of cMTs in the neck and exit from mitosis in cells with mispositioned spindles. As previously reported, we found that cMT loss from the bud neck does indeed precede exit from mitosis in cells that inappropriately breakdown their spindle in the mother cell compartment. However, our data show that loss of cMTs from the bud neck is not a cause but rather a consequence of exit from mitosis. We find, in cells wh exit from mitosis despite harboring a mispositioned spindle, that Cdc14 release from the nucleolus precedes rather than follows the disassembly of cytoplasmic microtubules and exit from mitosis. Second, we show that severing cytoplasmic microtubules does not lead to inappropriate exit from mitosis in cells with mispositioned spindles. Finally, we developed a method that allowed us to create cells containing two nuclei. We find that as long as one nucleus enters the bud during anaphase, cells will exit from mitosis, irrespective of whether the other nucleus is correctly or incorrectly positioned. Our data are inconsistent with a model where cMT-bud-neck interactions prevent exit from mitosis in cells with mispositioned spindles. Instead, they support the conclusion that spatial regulation of the MEN is controlled through the delivery of a MEN component bearing SPB into the bud.

## Results

### A system to monitor spindle misposition by live cell microscopy

Inactivation of either Kar9 or Dyn1 causes a fraction of cells to transiently misposition their spindles (*Figure 1C*). Such cells will then delay in anaphase until spindle position has been corrected (*Figure 1D*) (*Miller and Rose, 1998*). The relatively low penetrance and transient nature of the spindle positioning defect of *kar9Δ* and *dyn1Δ* cells has impeded the investigation of the consequences of spindle misposition on cell cycle progression. To overcome this limitation we developed a system to conditionally inactivate both the Kar9 and Dyn1 spindle positioning pathways. We depleted Dyn1 using the Indole-3-acetic acid (IAA; auxin) depletion system (*Nishimura et al., 2009*). IAA is a naturally occurring plant hormone that promotes the degradation of proteins containing an AID degron sequence by targeting them for ubiquitinylation by the SCF-Tir1 ubiquitin ligase (*Dharmasiri et al., 2005*; *Gray et al., 2001*; *Kepinski and Leyser, 2005*; *Teale et al., 2006*). We generated a strain carrying a *DYN1-AID* fusion and a deletion of *KAR9*, henceforth the *DYN1-AID kar9Δ* strain. Live cell imaging showed that 92% of *DYN1-AID kar9Δ* cells initially mispositioned their spindles (i.e. had spindles greater than 2 μm in length in the mother cell compartment) upon IAA addition in contrast to 13% seen in the wild type controls. This finding indicated that the *DYN1-AID kar9Δ* system effectively inactivates spindle-positioning systems in the cell.

To characterize the effects of spindle mispositioning on exit from mitosis we compared anaphase duration of cells with correctly aligned spindles to those with mispositioned spindles. Wild-type cells with correctly aligned spindles underwent anaphase within 19.2 ± 4.8 min (*Figure 1E and G*). In

contrast, *DYN1-AID kar9Δ* cells spent 65.9 ± 51.7 min in anaphase. This anaphase delay was highly variable. Most cells (85%) eventually were able to pull the mispositioned spindle into the bud (see *Figure 1H* for an example), which was followed by exit from mitosis. Only 6% of cells arrested with a mispositioned spindle for the duration of the movie analysis (longer than 200 min, open squares in *Figure 1E*). The fact that the majority of *DYN1-AID kar9Δ* cells eventually managed to correctly align their spindles along the mother – bud axis suggests that cells harbor residual dynein activity, perhaps because the depletion is not complete. It is also possible that additional minor spindle positioning pathways exist in these cells (*Kirchenbauer and Liakopoulos, 2013*; *Segal et al., 2002*).

Although exit from mitosis was prevented in the majority of cells with mispositioned spindles, we observed inappropriate exit from mitosis in 9% of such cells, leading to the formation of anucleate and binucleate cells (*Figure 1F and I*). This incomplete penetrance of the cell cycle arrest caused by spindle misposition has been observed previously (*Adames et al., 2001*; *D'Aquino et al., 2005*; *Pereira and Schiebel, 2005*). The reason why a small fraction of cells escapes the anaphase arrest caused by spindle misposition was, however, not understood.

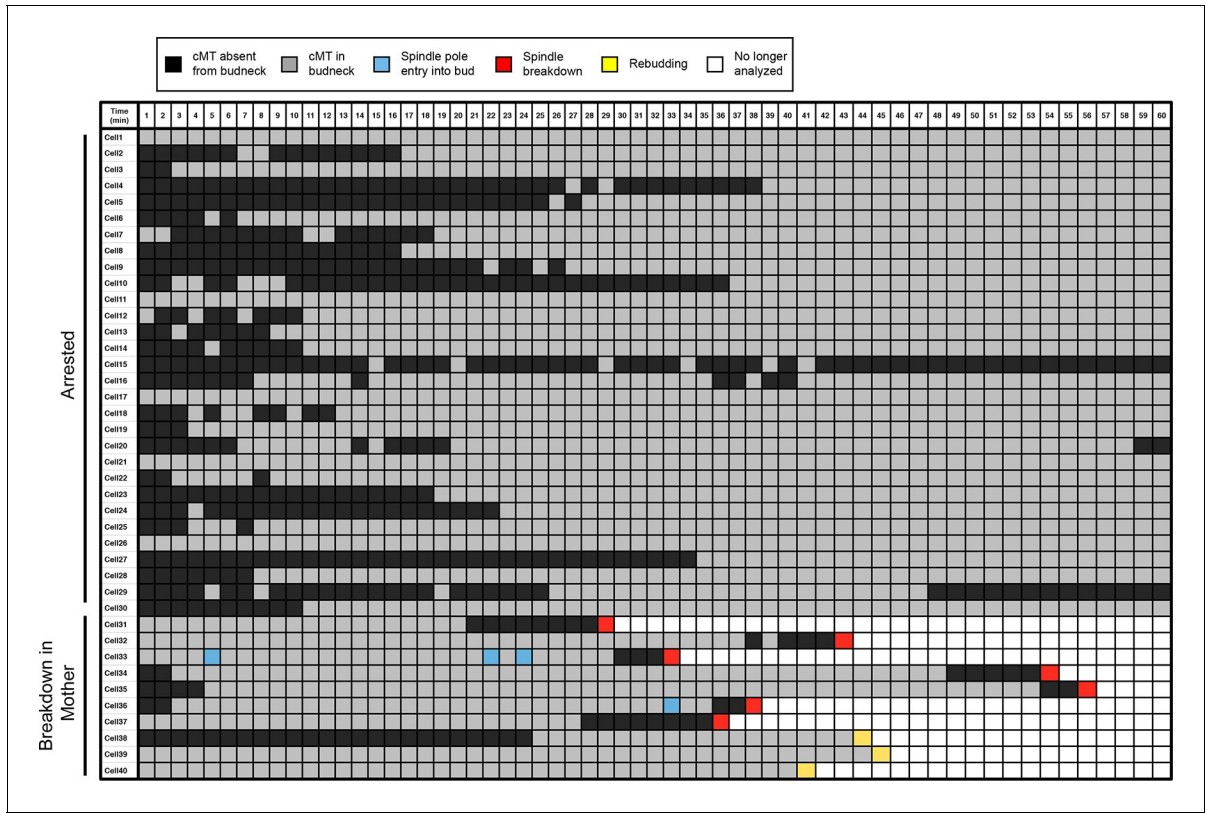

**Figure 2.** Analysis of cytoplasmic microtubules in the bud neck. Cells harboring *osTIR1 DYN-AID kar9Δ* and expressing GFP-labeled tubulin (A35707) were grown and imaged as descried in *Figure 1E–I*. A table summarizing cMT-bud neck contact for cells that contained a mispositioned spindle for 60-min (cells 1–30) or exited mitosis within that time frame (cells 31–40) is shown. Each row shows the color-coded fate of one cell for the given time period, as well as whether it had a cMT in contact with the bud neck. cMT analysis was performed by assessing whether a cMT was present or absent from the bud neck. Cells in which the tip of a cMT interacted with the bud neck or where the cMT traversed the bud neck was categorized as 'cMT in bud neck' (grey boxes) Cells lacking any cMT in the bud neck are described as 'cMT absent from bud neck' (black boxes). Movement of one spindle pole into the bud is described as 'spindle pole movement into bud' (blue boxes). Inappropriate exit from anaphase was determined by the spindle morphology and is described as 'spindle breakdown' (red boxes). A second category of inappropriate exit from mitosis was scored based on whether the cell rebudded without spindle collapse or cytokinesis (yellow boxes). Due to the low frequency of inappropriate spindle breakdown in the mother compartment, this table shows all cells that inappropriately exit mitosis from 2 experiments (cells 31–40). The cells that remain euploid (cells 1–30) are from experiment 1.

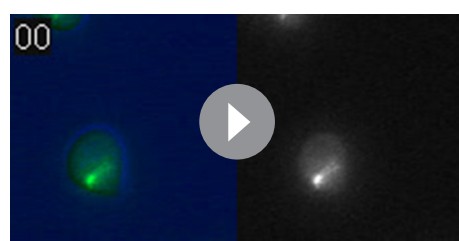

**Video 1.** A cell with a mispositioned spindle that has a cMT in the bud neck. *osTIR1 DYN-AID kar9Δ* cells expressing GFP-labeled α-tubulin (A35707) were grown and imaged as described in *Figure 1E-I*. Depicted is a cell that mis-positions its spindle and displays cMTs that protrude into the bud. Time in minutes is show in the upper left-hand corner of the time-lapse image. A merged image of the DIC and GFP channels is shown on the left. The GFP channel alone is shown on the right. The DIC image was adjusted to maximize contrast.

## Loss of cMT- bud neck contacts does not cause inappropriate mitotic exit in cells with mispositioned spindles

One hallmark of cells with mispositioned spindles is long cMTs that emanate from one or both SPBs through the bud neck and into the bud (*Figure 1H*; 100 min time point). In the small fraction of cells that escape the anaphase arrest caused by spindle misposition, inappropriate mitotic exit is preceded by retraction of cytoplasmic microtubules from the bud neck (*Moore et al., 2009*). Having established a tool to induce spindle misposition in many cells, we decided to reinvestigate this correlation.

As reported previously, we found that cells with mispositioned spindles frequently display cMTs that contact the bud neck (*Figure 2*, cells 1–30; *Video 1*). Also consistent with previous studies, we found that in cells that exit from mitosis despite harboring a mispositioned spindle, this cell cycle transition was preceded by the loss of cMT-bud neck interactions. Contact was lost approximately 2–8 min before spindle breakdown (*Figure 2*, cells 31–37; *Video 2*). Additionally, we found a small number of cells that did not lose cMT-bud neck contact but entered the next cell cycle as assessed by budding (*Figure 2*, cells 38–40; *Video 3*). These cells also did not completely breakdown their spindle and did not complete cytokinesis. Importantly, our analysis also revealed that inappropriate mitotic exit was not an obligatory consequence of loss of cMT – bud neck interactions. The majority of cells with mispositioned spindles lacked cMT-bud neck contacts for significant periods of time yet stayed arrested in anaphase (*Video 4*; *Figure 2*, cells 1–30). We conclude that loss of cMT – bud neck interactions does not necessarily cause inappropriate mitotic exit in cells with mispositioned spindles.

A previous study reported that eliminating cMT-bud neck interactions by ablating GFP-labeled microtubules causes exit from mitosis in cells with mispositioned spindles (*Moore et al., 2009*). We conducted a similar analysis and found this not to be the case. We used a laser to ablate microtubules in cells either lacking both the *KAR9* and *DYN1* spindle positioning pathways or just the *DYN1* positioning pathway. We ablated the cMT that interacted with the bud neck in 15 *DYN1-AID kar9Δ* cells, but exit from mitosis was not observed in the mother cell compartment within the time that we monitored cells post severing (69 min; *Figure 3C*). We also assessed exit from mitosis following laser ablation in these cells using a marker for cytokinesis. None of the 10 cells with mispositioned spindles in which we ablated the bud-neck interacting cMT exited mitosis as judged by loss of septin Cdc3 from the bud neck (*Figure 3A and C*). To ensure that exposure of cells to the laser pulse did not cause cell cycle arrest, we targeted the cytoplasm of cells with correctly positioned spindles with the laser. In the 20 cells treated in this manner, exit from mitosis occurred within 2–24 min following application of the laser pulse (*Figure 3B; C*, aligned category). Furthermore, when laser

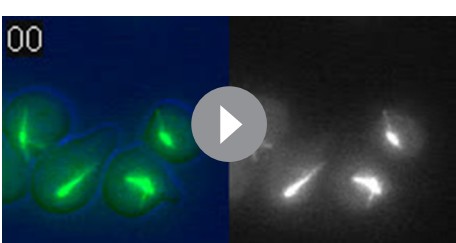

**Video 2.** A cell with a mispositioned spindle that inappropriately exits mitosis in the mother cell compartment. *osTIR1 DYN-AID kar9Δ* cells expressing GFP-labeled α-tubulin (A35707) were grown and imaged as described in *Figure 1E-I*. Depicted is a cell that mis-positions its spindle and then inappropriately exits mitosis in the mother cell. Time in minutes is show in the upper left-hand corner of the time-lapse image. A merged image of the DIC and GFP channels is shown on the left. The GFP channel alone is shown on the right. The DIC image was adjusted to maximize contrast.

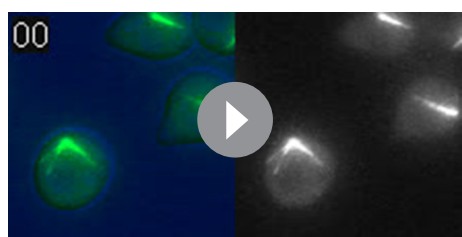

**Video 3.** A cell with a mispositioned spindle that inappropriately exits mitosis but does not complete spindle disassembly or cytokinesis. *osTIR1 DYN-AID kar9Δ* cells expressing GFP-labeled α-tubulin (A35707) were grown and imaged as described in *Figure 1E-I*. Depicted is a cell that mispositions its spindle and then inappropriately exits mitosis in the mother cell but does not complete cytokinesis or spindle disassembly. Time in minutes is show in the upper left-hand corner of the time-lapse image. A merged image of the DIC and GFP channels is shown on the left. The GFP channel alone is shown on the right. The DIC image was adjusted to maximize contrast.

ablated cells with mispositioned spindles managed to realign their spindle, they also exited mitosis (*Figure 3C*, spindle breakdown in the bud category). Lastly, we ablated cMTs in *dyn1Δ* cells. Exit from mitosis did not occur for the duration of the analysis (69 min) in cells with mispositioned spindles in which cMTs were ablated. One cell succeeded in positioning its spindle correctly along the mother – bud axis and promptly exited mitosis thereafter. In 2 cells exit from mitosis followed ablation of cMTs. One cell exited mitosis 8 min post ablation and the other cell exited mitosis 69 min post ablation. The latter instance seems unlikely to be the consequence of loss of cMTs from the budneck because the cell initially lacked cMTs in the budneck for at least 7 min but exited mitosis much later (69 min post-ablation). The one cell that exited mitosis shortly after cMT ablation is well in line with the fraction of wild-type cells that exit from mitosis despite harboring a misaligned spindle (*Figure 1F*). In summary, our results show that ablation of cMTs does not promote exit from mitosis in the vast majority (33/35) of cells analyzed (*Figure 3C*, mispositioned spindles category). We conclude that although cMT retraction frequently precedes inappropriate exit from mitosis in cells with misaligned spindles, it is not the cause of exit from mitosis in these cells.

## Cdc14 release from the nucleolus precedes loss of cMT – bud neck interactions in cells that exit mitosis despite containing a mispositioned spindle

If loss of cMT-bud neck interactions does not induce inappropriate exit from mitosis in cells with misaligned spindles, what does? To begin to address this question we asked whether inappropriate exit from mitosis in cells with mispositioned spindles relied on the same regulatory pathways that promote exit from mitosis in cells with correctly positioned spindles.

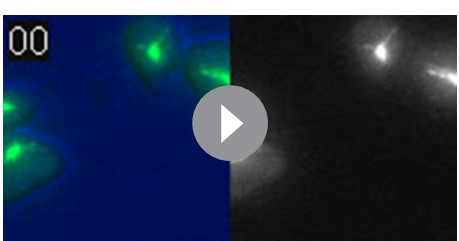

**Video 4.** A cell with a mispositioned spindle that lacks cMT-bud neck interactions but arrests in anaphase. *osTIR1 DYN-AID kar9Δ* cells expressing GFP-labeled α-tubulin (A35707) were grown and imaged as described in *Figure 1E-I*. Depicted is a cell that arrests in late anaphase with a mispositioned spindle. This cell lacks cMTs in the bud neck for a substantial time period in anaphase. Time in minutes is show in the upper left-hand corner of the time- lapse image. A merged image of the DIC and GFP channels is shown on the left. The GFP channel alone is shown on the right. The DIC image was adjusted to maximize contrast.

Cdc14 is the key trigger of exit from mitosis (reviewed in *Stegmeier and Amon [2004]*). Cdc14 release from the nucleolus during anaphase activates the phosphatase to trigger exit from mitosis. We used Cdc14 localization as the criterion to determine whether cMT retraction from the bud neck occurred before or after exit from mitosis. To examine Cdc14 localization we used a Cdc14-tdTomato fusion. This allele is slightly hypermorphic (it causes inappropriate exit from mitosis in a small fraction of cells with mispositioned spindles after a substantial anaphase delay, *Figure 4—figure supplement 1A*) but nevertheless accurately reflects the changes in Cdc14 localization during the cell cycle (*Figure 4—figure supplement 1B*).

Live cell imaging showed that Cdc14 release from the nucleolus preceded both, cMT retraction from the bud neck and mitotic spindle breakdown in cells that exited mitosis despite harboring a misaligned spindle. In 78.22 ± 3.0% of cells (n ≥ 37 cells per biological replicate. 3

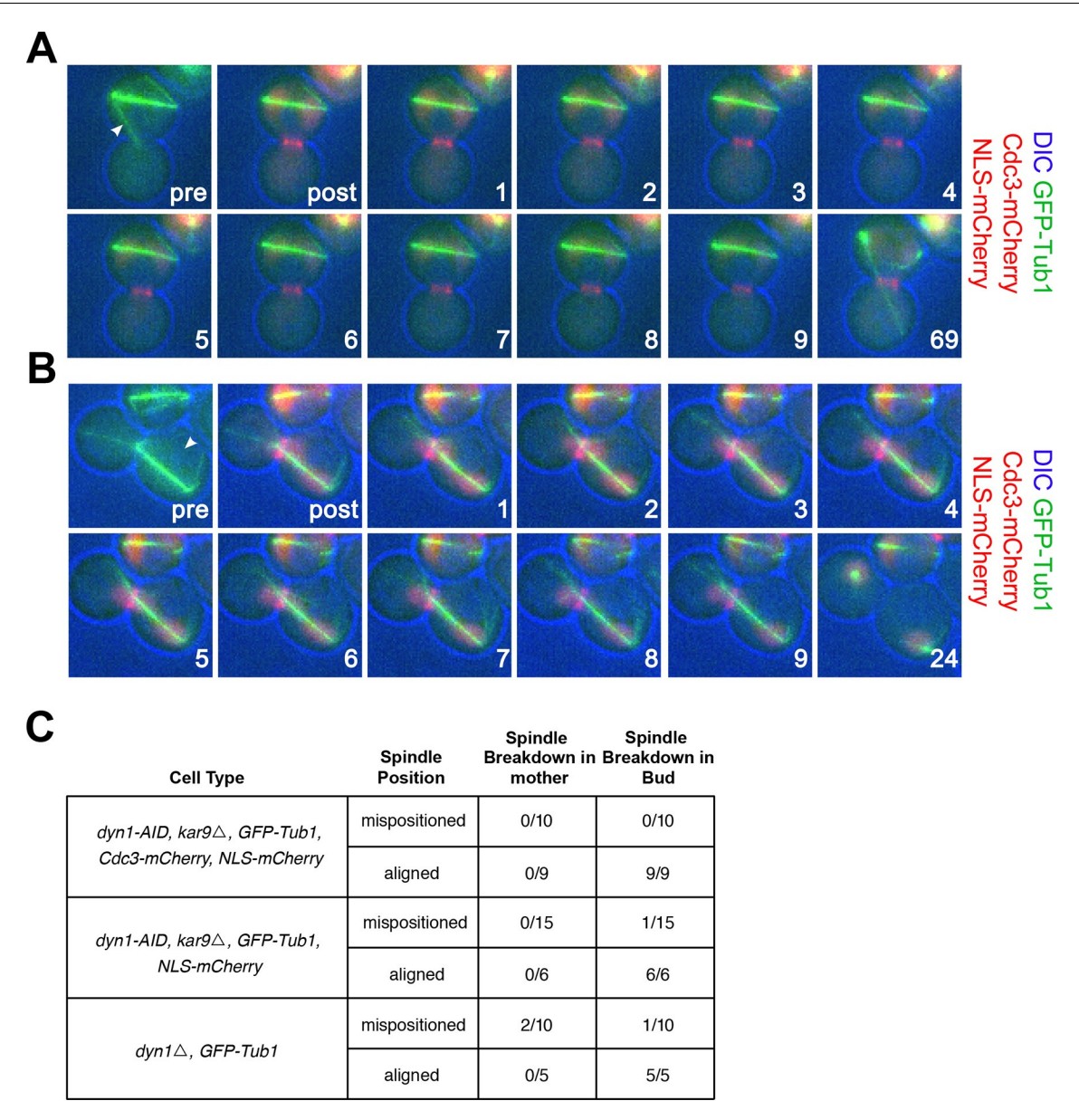

**Figure 3.** cMT laser ablation does not promote exit from mitosis in cells with mispositioned spindles. (**A-B**) Cells harboring the *osTIR1 DYN-AID kar9Δ* constructs that also expressed GFP-tagged α-tubulin, mCherry-tagged Cdc3 and an NLS-mCherry (A35143) were grown overnight to mid-log in YEPD at 25°C. Cells were then resuspended in synthetic complete medium supplemented with 100 μM IAA and incubated for 2–3 hr at 25°C. The cells were prepared on an agar pad for live cell microscopy. Two pre-ablation images were taken (only one is shown) before the cMT was cut. Post ablation cells were monitored for 9 min at 1-min intervals for cMTs and then 1 hr at 15-min intervals to follow cell cycle progression. The arrowheads indicate the approximate laser targeting site. (**A**) A montage of a cell with a mispositioned spindle where cMT bud neck interactions were disrupted due to microtubule severing. The DIC channel is optimized to enhance contrast. (**B**) A montage of a control cell with an aligned spindle where the laser was targeted to the cytoplasm. The DIC channel is optimized to enhance contrast. (**C**) Table summarizing cell cycle stage of aligned and misaligned spindles 69 min post ablation. The culturing conditions for cells in the first (A35143) and second rows (A34832: the same as A35143 but lacking Cdc3-mCherry) of the table are the same as described above. In the third row, cells lacking *DYN1* and expressing GFP-tagged α-tubulin and NLS-mCherry (A34722) were grown overnight at 25°C to mid-log and then shifted to 16°C for 2–5 hr to enrich for cells with mispositioned spindles. The cells were then mounted on an agar pad for live cell microscopy. One set of pre-ablation images was taken before the cMT was cut. Post ablation cells were monitored as described above.

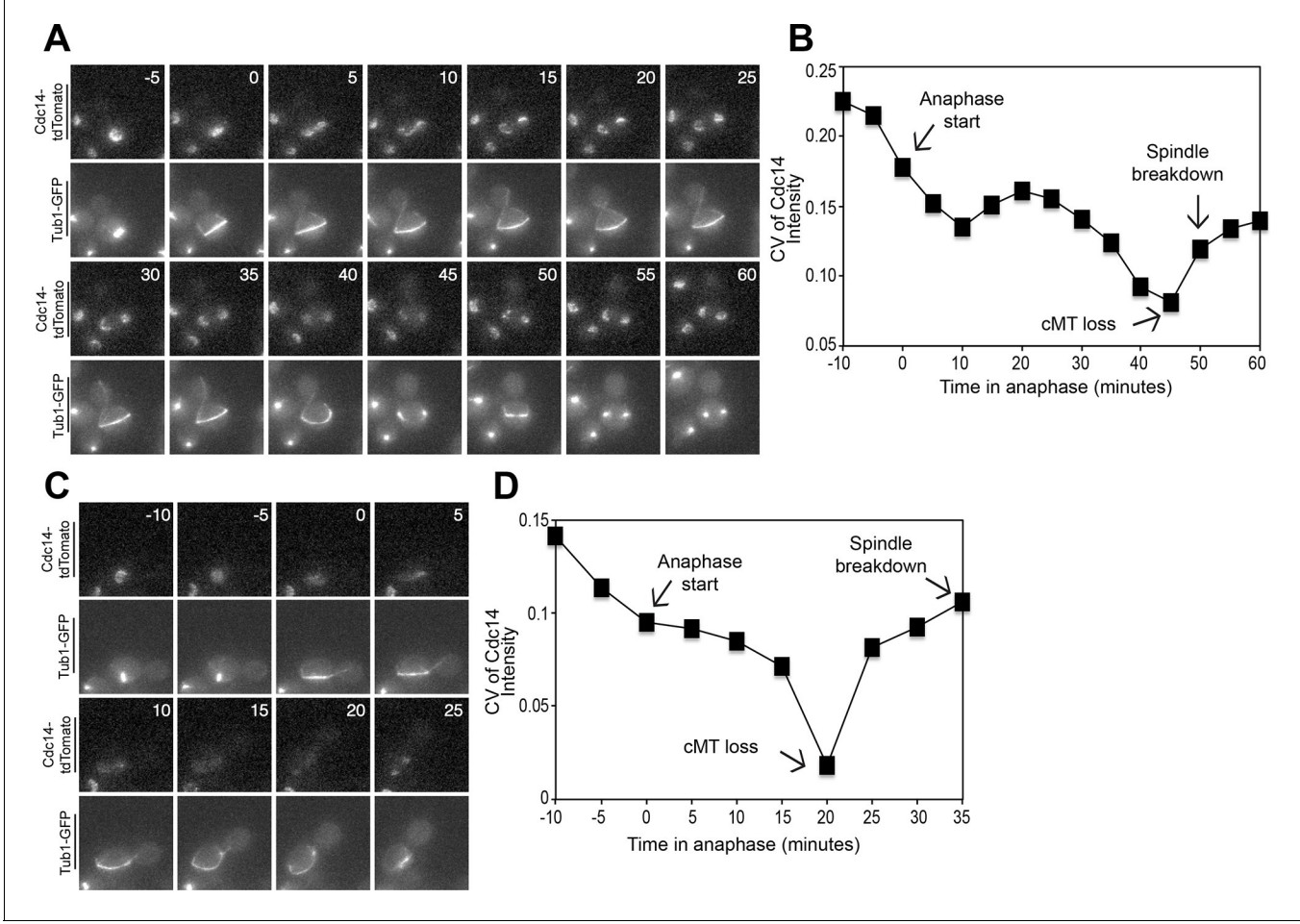

**Figure 4.** Cdc14 Release from the nucleolus precedes cMT retraction in cells that inappropriately breakdown their anaphase spindles in the mother cell compartment. (A–D) A diploid strain (A37463) homozygous for *osTIR1 dyn1-AID kar9Δ* and heterozygous for GFP-Tub1 and Cdc14-tdTomato was grown to midlog in Synthetic Complete medium. Cycling cells were imaged on a flow cell in Synthetic Complete medium supplemented with 100 μM IAA. (A) Representative images are shown for the anaphase release of Cdc14-tdTomato with respect to GFP-Tub1 cMT retraction. (B) The coefficient of variation (the standard deviation divided by the mean) was measured for Cdc14 pixel intensity for the cell pictured in the *Figure 4A* montage. Time (in minutes) is displayed on the X-axis and the zero time point reflects anaphase onset. (C) Representative images are shown for a cell with a mispositioned spindle in which complete Cdc14-td Tomato release from the nucleolus was observed at anaphase onset. cMTs are also shown for that cell during the same time interval. (D) The coefficient of variation (the standard deviation divided by the mean) was measured for Cdc14 pixel intensity for the cell pictured in the *Figure 4C* montage. Time (in minutes) is displayed on the X-axis. The zero time point reflects anaphase onset. See also: *Figure 4—figure supplement 1–4*.

The following figure supplements are available for figure 4:

**Figure supplement 1.** Analysis of the *CDC14-tdTomato* allele.

**Figure supplement 2.** Analysis of Cdc14 Release in cells with mis-positioned spindles.

**Figure supplement 3.** Montage of Cdc14 Release in cells with mis-positioned spindles containing the nucleolar marker Cfi1.

**Figure supplement 4.** Montage of Cdc14 Release in cells with mis-positioned spindles containing the nucleolar marker Cfi1.

replicates were performed), Cdc14 release occurred approximately 5 min prior to loss of cMT – bud neck interactions and approximately 5–15 min prior to mitotic spindle breakdown (*Figure 4A* and *Figure 4—figure supplement 2*). The decrease in nucleolar Cdc14 signal intensity that occurred shortly prior to spindle breakdown was not caused by changes in nucleolar morphology that take

place during anaphase. Signal intensity of Cdc14's nucleolar anchor Cfi1/Net1 did not change during anaphase (*Figure 4—figure supplements 3* and *4*). Instead, it appears that Cdc14 was released from the nucleolus. In the majority of cells (73.85 ± 4.4%) Cdc14 release from the nucleolus occurred during anaphase (*Figure 4A* and *Figure 4—figure supplement 2A*). However in a small fraction of these cells, 4.4 ± 4.1%, Cdc14 appeared to be fully released already at the metaphase to anaphase transition (*Figure 4C*).

To more precisely analyze when Cdc14 was released from the nucleolus relative to cMT retraction, we calculated the coefficient of variation (CV; standard deviation/mean) of Cdc14-tdTomato pixel intensity in the whole cell over time (*Lu and Cross, 2010*). As Cdc14 is released from the nucleolus and spreads throughout the nucleus and later the cytoplasm, the standard deviation of pixel intensities will decrease as cells progress through anaphase. We should note that despite intense efforts, we were not able to normalize changes in Cdc14-tdTomato CV to that of a protein that localizes to the nucleolus in a constitutive manner. GFP-tagged nucleolar markers overlapped with the Tubulin-GFP signal, a construct that was necessary to determine when cMTs retract. Tags other than GFP, such as BFP, were too dim to detect by live cell microscopy. Despite this limitation, it was nevertheless clear that the coefficient of variation of Cdc14-td Tomato pixel intensity decreased before cMT retraction from the bud neck (*Figure 4B,D*, *Figure 4—figure supplement 2B*). Not all cells showed release of Cdc14 from the nucleolus prior to cMT retraction from the bud neck. In 10.8 ± 4.9% of cells Cdc14 release from the nucleolus occurred concomitantly with cMT retraction. In the remaining cells (11.0 ± 2.2%) that did not release Cdc14 prior to cMT retraction, Cdc14 release from the nucleolus was not detected prior to anaphase spindle breakdown. This is most likely because the fraction of Cdc14 that was released from the nucleolus was too small to detect by imaging. Importantly, we never observed that Cdc14 release occurred after cMT retraction from the bud neck. Consistent with the idea that Cdc14 triggers inappropriate exit from mitosis in cells with mispositioned spindles we find that depletion of Cdc14 suppressed exit from mitosis in the rare wild-type cells that escape the anaphase arrest when their spindles are mispositioned (*Figure 5A*). We conclude that inappropriate mitotic exit in cells with mispositioned spindles is not due to loss of cMT – bud neck interactions. Rather, cMT retraction is a consequence of Cdc14 release from the nucleolus in these cells. We further propose that the reason why loss of cMT – bud neck interactions always precedes mitotic spindle breakdown in cells with mispositioned spindles that exit from mitosis is because disassembly of a single cMT occurs more quickly than disassembly of the mitotic spindle.

## The FEAR network is required for Cdc14 release from the nucleolus in cells that exit from mitosis despite a mispositioned spindle

Two signaling pathways control Cdc14 release from the nucleolus. The FEAR network promotes the transient release of the phosphatase from the nucleolus during early anaphase. The MEN is needed for the sustained release of the phosphatase during later stages of anaphase (*Stegmeier and Amon, 2004*). Our data showing that Cdc14 is inappropriately released in some cells with mispositioned spindles led us to investigate which pathway regulating Cdc14 was responsible for promoting inappropriate Cdc14 release in these cells. Not surprisingly, inhibition of the MEN, a pathway which is essential for exit from mitosis in all cells, suppressed the inappropriate mitotic exit that is observed in the 11% of cells that exit from mitosis when their spindle is mispositioned (*Figure 5B*). We were, however, surprised to find that inactivation of the FEAR network, either by deleting *SPO12* or *SLK19* also prevented the inappropriate exit from mitosis in cells with mispositioned spindles (*Figure 5C*) (*Scarfone et al., 2015*). Consistent with the idea that cMT – bud neck interactions are not regulating mitotic exit, we found that deleting *SPO12* largely did not affect these interactions (compare *Figures 2* and *5E*) despite completely suppressing inappropriate mitotic exit in cells with mispositioned spindles. Finally, in cells with mispositioned spindles, we found that the release of Cdc14 from the nucleolus was prevented in cells lacking *SPO12* (*Figure 5D*).

Together, our data lead to the following two conclusions. First, cMT – bud neck interactions are not responsible for preventing Cdc14 activation and exit from mitosis in response to mispositioned spindles. Instead, activation of Cdc14 causes cMT retraction from the bud neck and exit from mitosis in cells with mispositioned spindles that inappropriately exit from mitosis. Second, Cdc14 activation in the cells that exit from mitosis despite harboring a mispositioned spindle depends on the FEAR network. This observation raises the interesting possibility that it is high FEAR network activity that causes bypass of the anaphase arrest triggered by spindle misposition in some cells.

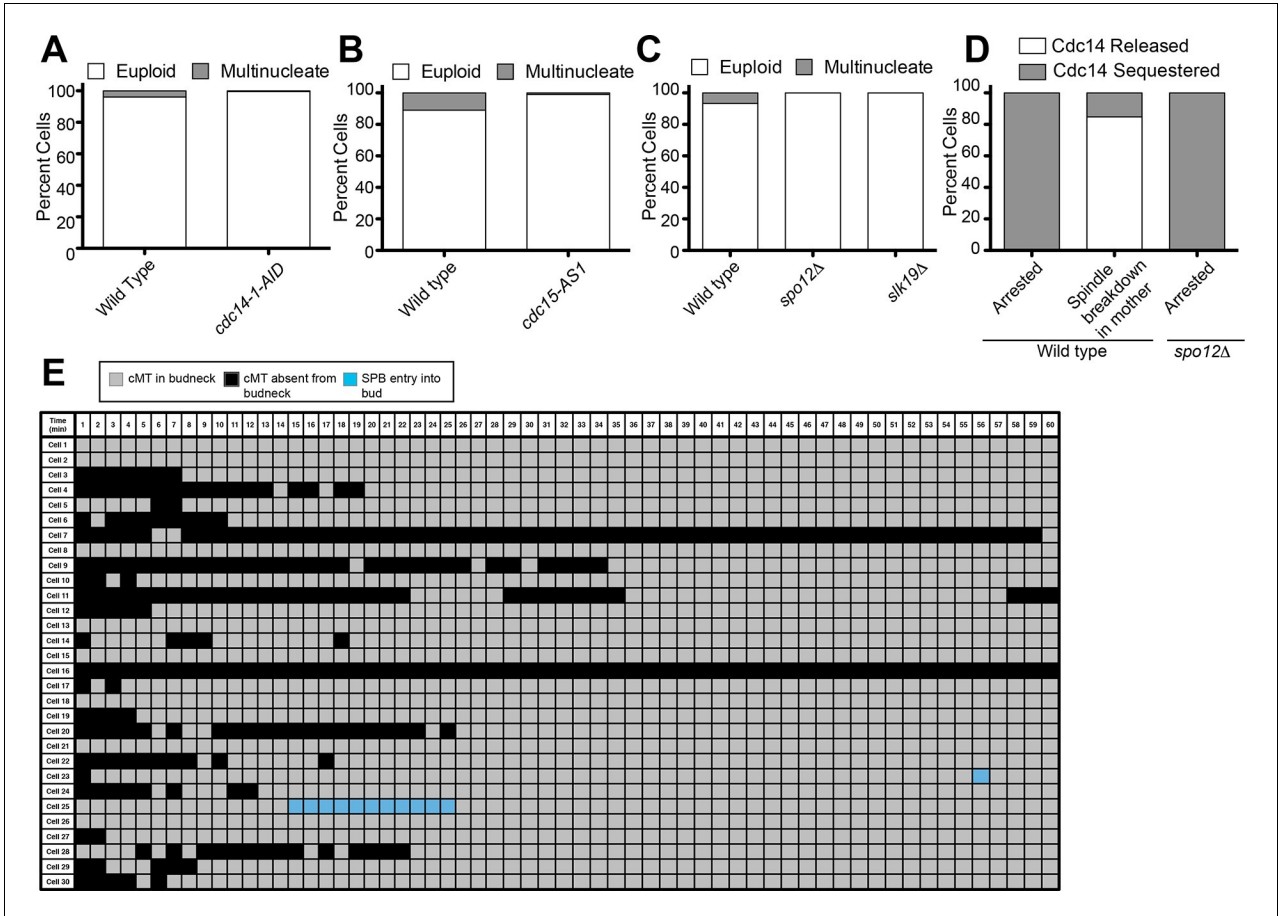

**Figure 5.** Inhibition of Cdc14, the MEN and the FEAR Network prevents Spindle Breakdown in the mother cell compartment. (A) *osTIR1 dyn1-AID kar9Δ* (A35707) and *osTIR1 dyn1-AID kar9Δ cdc14-1-AID* (A37895) cells harboring GFP-tagged α-tubulin were grown as described in *Figure 1E–I*. Cells were monitored by live cell microscopy and scored for inappropriate spindle breakdown in the mother cell compartment. n=100 for *osTIR1 dyn1-AID kar9Δ* and 225 for *osTIR1 dyn1-AID kar9Δ cdc14-1-AID*. (B) *osTIR1 dyn1-AID kar9Δ* (A35707), *osTIR1 dyn1-AID kar9Δ cdc15-AS1* (A36264) expressing GFP-tagged α-tubulin, were analyzed as in *Figure 5A* but with the addition of 20 µM NAPP1 to the medium. Cells were imaged in a Lab-Tek II chamber. n=100 for each strain. (C) *osTIR1 dyn1-AID kar9Δ* (A35707), *osTIR1 dyn1-AID kar9Δ spo12Δ* (A35700), *osTIR1 dyn1-AID kar9Δ slk19Δ* (A36028) expressing GFP-tagged α-tubulin, were analyzed as in *Figure 5A*. n ≥ 284 for each strain. (D) Wild type (A37753) or *spo12Δ* (A37610) diploid strains homozygous for *osTIR1 dyn1-AID kar9Δ and GFP-Tub1* and heterozygous for Cdc14-tdTomato were grown to mid log in synthetic complete medium. Cells with mispositioned spindles that had two distinct nucleoli were scored based on whether Cdc14-tdTomato was released from the nucleolus in late anaphase. Cells that did not exit mitosis in the mother cell compartment were monitored for 60 min. n≥22 cells. (E) cMT analysis was performed on *osTIR1 dyn1-AID kar9Δ spo12Δ* (A35700) as described in *Figure 2*. Each row shows the color-coded fate of one cell for the given time period, as well whether it had a cMT in contact with the bud neck. cMT analysis was performed by assessing whether a cMT was present or absent from the bud neck. Cells with a cMT end that was in the bud neck or in the bud was categorized as 'cMT in bud neck' (grey boxes) Cells lacking any cMT in the bud neck are described as 'cMT absent from bud neck' (black boxes). Movement of one spindle pole into the bud is represented by the 'spindle pole movement into bud category (blue boxes). Note: Cell #25 transiently moves a spindle pole into the bud cell compartment but does not exit from mitosis. This is most likely due the inefficient activation of the MEN in cells lacking the FEAR network.

## Spindle elongation into the bud signals correct spindle position and triggers exit from mitosis

A central tenet of the cMT - budneck model is the prediction that as long as a spindle is mispositioned, mitotic exit is inhibited. The 'zone model' or any other model that posits a mitotic exit-activating signal in the bud predicts the opposite. As long as a spindle is correctly positioned along the mother – bud axis exit from mitosis will occur, even if the cell also harbors a spindle that is mispositioned.

To determine whether a MEN inhibitory signal caused by a misaligned spindle prevents MEN activation and hence exit from mitosis or whether a correctly aligned spindle activates the MEN and

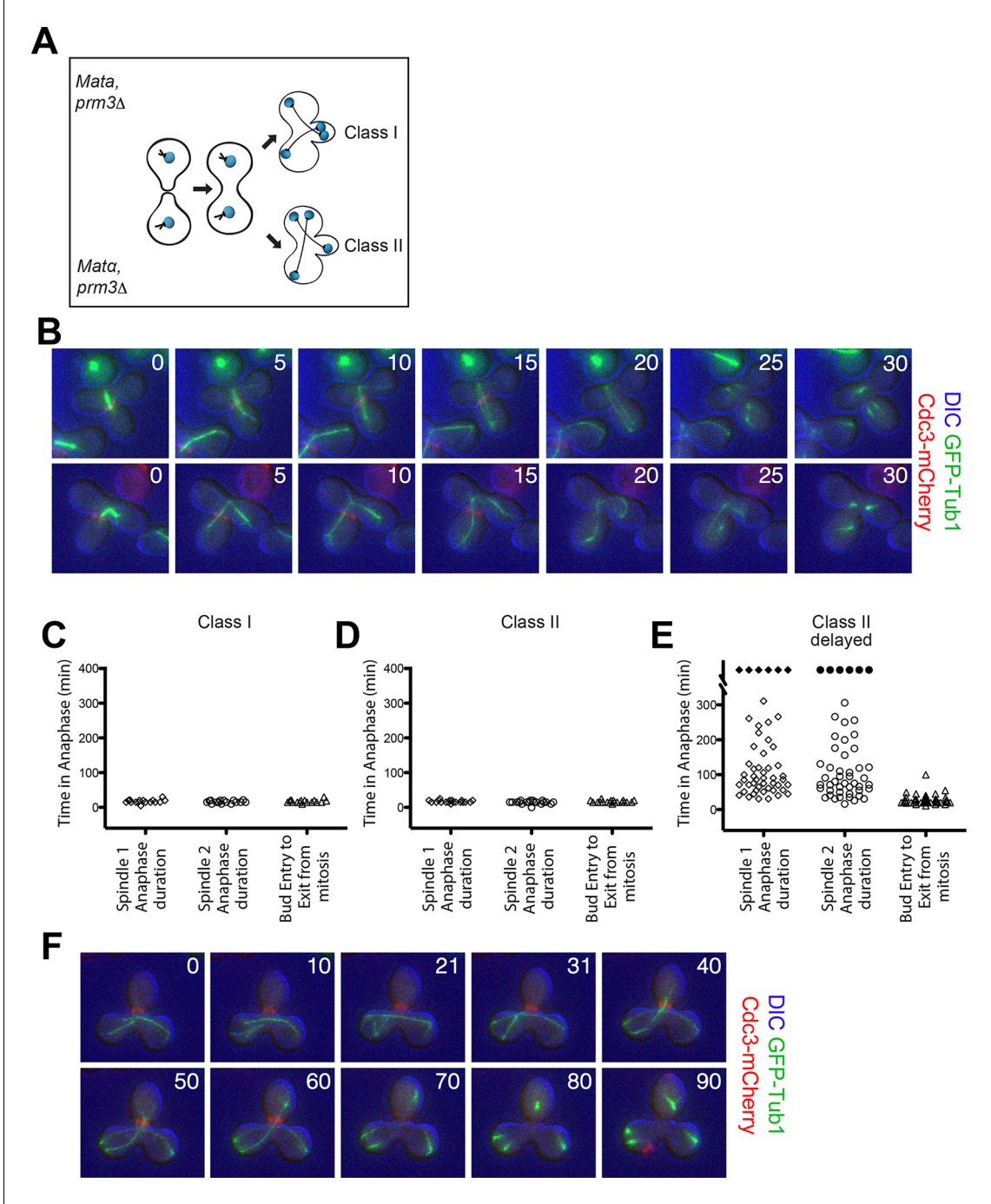

**Figure 6.** Analysis of Exit from Mitosis in *prm3Δ* heterokaryons. (**A**) Cartoon of the *prm3Δ* heterokaryon system showing the two main classes of heterokaryons obtained. (**B–F**) Heterokaryons were obtained by mating cells lacking *PRM3* (see Materials and methods for details). Briefly, G1 cells isolated by centrifugal, elutriation were mated at 30° for 2 hr and then loaded onto a Y04D CellASIC flow cell for imaging. Synthetic complete pH 6.0 medium was used in the flow cell during imaging and was supplemented with 100 μM IAA to induce spindle mispositioning. (**B**) Montages of binucleate zygotes created by mating cells lacking *PRM3*. Binucleate diploids are homozygous for *prm3Δ* and *osTIR1* and heterozygous for *DYN-AID, kar9Δ*, mCherry-labeled Cdc3 and GFP-labeled α-tubulin (A37892 x A35570). The DIC channel was adjusted to maximize contrast. (**C–D**) Analysis of anaphase kinetics of cells described in *Figure 6B*. Class 1: n= 16. Class II: n= 21 (**E**) Binucleate diploids that were homozygous for *prm3Δ, osTIR1, DYN-AID, kar9Δ* and heterozygous for mCherry-labeled Cdc3 and GFP-labeled α-tubulin (A35570 x A35571) were analyzed for anaphase duration. Black diamonds indicated permanently arrested cells (permanently arrested ≥320 min) n=51 cells. (**F**) Montage of cells from *Figure 6E*. The DIC channel was adjusted to maximize contrast.

hence triggers mitotic exit, we generated cells with two nuclei (henceforth, heterokaryons). When these cells undergo anaphase two main classes of cells are observed (*Figure 6A,B*):

Class I cells: both spindles correctly align along the mother (*Figure 6A,B* - top panel).

Class II cells: one spindle aligns whereas the other one is misaligned (*Figure 6A,B* - bottom panel).

We generated heterokaryons using two different methods. In the first method, we created cells with two nuclei by mating two cells that lacked the nuclear fusion gene *PRM3* (*Figure 6A*) (*Heiman and Walter, 2000*; *Shen et al., 2009*). In cells in which both spindles were correctly aligned along the mother – bud axis (Class I cells) both spindles entered the bud in quick sequence during anaphase and exit from mitosis (as judged by Cdc3 loss from the bud neck) occurred promptly thereafter (*Figure 6C*). Class II cells also exited mitosis even though only one spindle entered the bud during anaphase. Average anaphase duration for these cells was 15.7 ± 2.3 min, which was comparable to the average anaphase duration in cells in which both spindles were correctly positioned (16.6 ± 3.7 min; compare *Figure 6C,D*). Furthermore the time of entry of one spindle into the bud until exit from mitosis was the same in cells in which both spindles aligned correctly and cells in which one spindle was correctly aligned and the other was misaligned (compare *Figure 6C,D* column 'bud entry to exit from mitosis').

Importantly, we were also able to obtain many heterokaryons where both anaphase spindles were mispositioned for prolonged periods of time (henceforth 'Class II delayed'). These cells were severely delayed in anaphase (*Figure 6E,F*). In the vast majority of these cells it was only once one spindle moved from the mother compartment into the bud that cells promptly exited mitosis (42/51 cells). Of the remaining cells, six cells (11.76%) permanently arrested in anaphase with both spindles in the mother cell compartment and two inappropriately exited mitosis with both spindles in the mother cell (3.92%). Strikingly, we also noticed that in a very small fraction of cells (1/51), exit from mitosis occurred even when the mispositioned spindle in the mother cell had not yet initiated anaphase (*Figure 7*). We conclude that movement of one spindle pole into the bud triggers exit from mitosis.

The second method to generate heterokaryons took advantage of the fact that cells undergoing meiosis can be returned to vegetative growth (*Simchen, 2009*). Diploid yeast cells sporulate in response to nutrient deprivation. Budding is suppressed and cells progress through premeiotic S-phase. When glucose-containing medium is supplied to cells that lack the CDK inhibitory kinase Swe1 following premeiotic S phase (in pachytene of meiotic prophase I) these cells will return to vegetative growth and undergo mitosis producing cells with two nuclei (*Figure 8A*)(*Tsuchiya and Lacefield, 2013*). We analyzed the mitotic cell cycle after the formation of a binucleate cell. In this method of generating heterokaryons, a third class of cells was observed: one spindle is pulled into the bud generating cells with a misaligned spindle in the mother and bud cell compartments (*Figure 8A*). As in the heterokaryons generated by mating, cells in which both spindles were

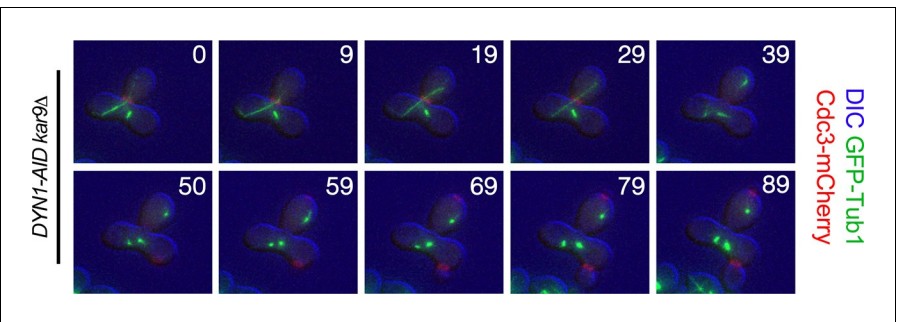

**Figure 7.** Exit from Mitosis in a *prm3Δ* heterokaryon with one aligned anaphase spindle and one metaphase spindle. Montage of binucleate zygotes created by mating homozygous for *prm3Δ, osTIR1, DYN-AID, kar9Δ* and heterozygous for mCherry-labeled Cdc3 and GFP-labeled α-tubulin (A35570 x A35571). The montage depicts a zygote with one aligned anaphase spindle and a second spindle in metaphase in the mother compartment. Both spindles exit mitosis at the same time with the metaphase spindle never going through anaphase. The DIC channel was adjusted to maximize contrast.

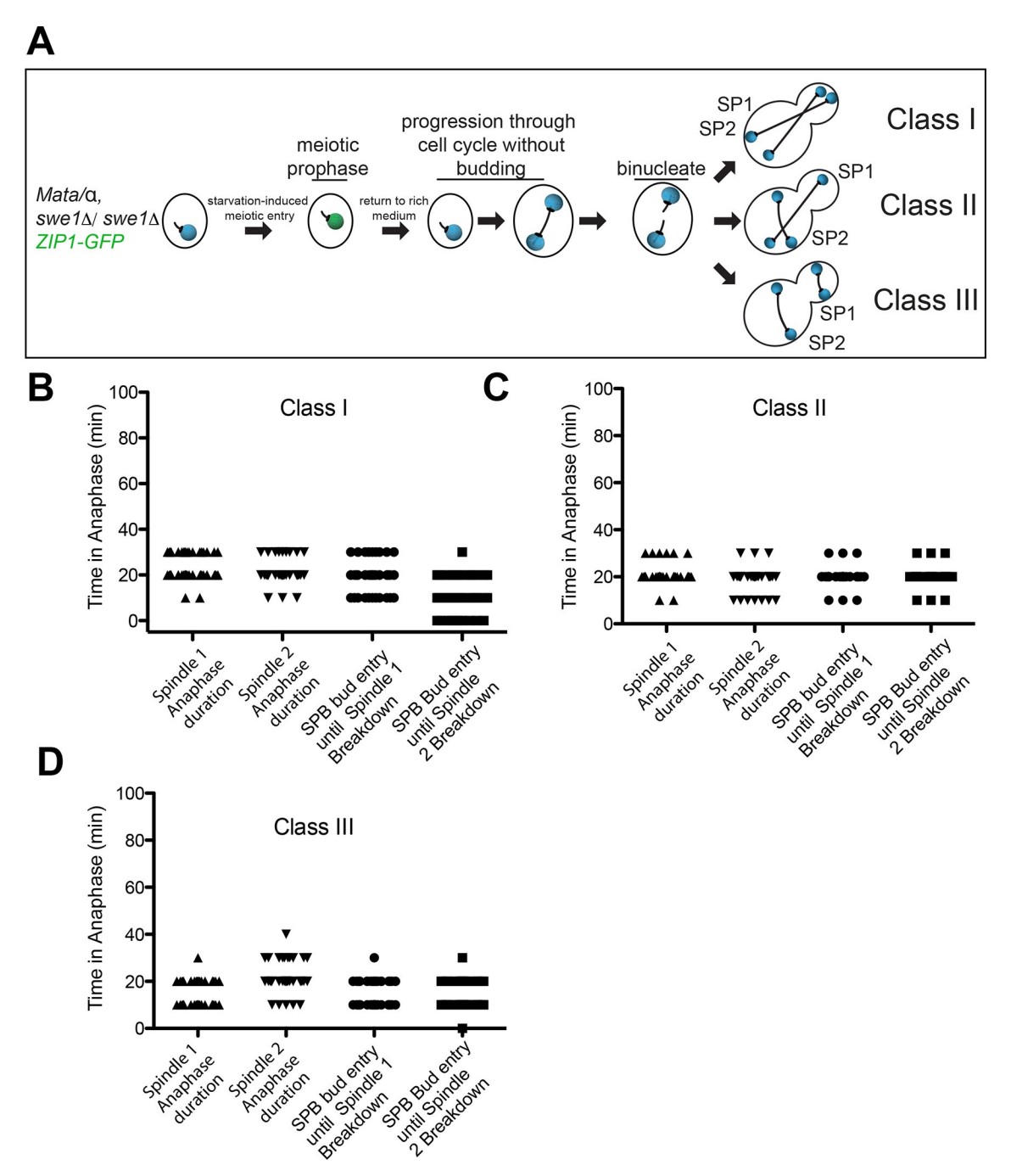

**Figure 8.** Analysis of Exit from Mitosis in swe1Δ heterokaryons. (**A**) Cartoon of *swe1Δ* return to growth heterokaryon system and depictions of cell type classes that were analyzed. Briefly, diploid cells lacking *SWE1*, also harboring the meiotic prophase marker Zip1, tagged with GFP as well as GFP-tagged Tub1 (LY1043), were induced to enter meiotic prophase through nutrient starvation. Upon entry into meiotic prophase (as judged by the presence of Zip1-GFP positive cells), cells were returned to glucose-containing complete medium in microfluidic chambers and thus induced to grow mitotically. These cells were monitored by live cell microscopy. (**B**) Anaphase kinetics of Class I cells (depicted in *Figure 8A*). Anaphase duration is classified as the time the cell spends with a spindle >2 μm to spindle breakdown. 'Bud entry to exit from mitosis' is defined as the time from when at least one spindle pole is in the bud to anaphase spindle disassembly. (**C**) Anaphase kinetics of Class II cells (depicted in *Figure 8A*). Anaphase duration and bud entry to exit from mitosis are define as in *Figure 8B*. (**D**) Anaphase kinetics of Class III cells (depicted in *Figure 8A*). Anaphase duration and bud entry to exit from mitosis are define as in *Figure 8B*. n=50 cells for each class.

correctly aligned along the mother – bud axis (Class I cells) exited from mitosis as judged by the average anaphase spindle breakdown 15.8 ± 5.5 min after the two spindles had entered the bud (*Figure 8B*). Class II cells also exited mitosis even though only one spindle entered the bud during anaphase. Anaphase duration was similar for both spindles and both spindles broke down soon after one spindle entered the bud (*Figure 8C*).

Spindle disassembly of the nucleus in the mother cell also occurred concomitantly with spindle disassembly in the nucleus in the bud of class III cells (*Figure 8D*). This result indicates that exit from mitosis is not triggered by a correctly positioned spindle but rather a spindle that is in the bud, as exit from mitosis occurred in class III cells even though the spindle in the bud was mispositioned. It is important to note however, that a spindle simply being in the bud is not sufficient to bring about exit from mitosis. Exit from mitosis only occurred after the bud-localized spindle had undergone anaphase. This observation is consistent with previous findings showing that anaphase entry is required for MEN activation and exit from mitosis (*Rock and Amon, 2011*) (data not shown).

In summary, our heterokaryon analyses do not support the hypothesis that a dominant inhibitory signal originating from a mispositioned spindle prevents MEN activation and exit from mitosis. Instead, our data show that movement of the spindle into the bud as occurs during a cell cycle with a correctly positioned spindle activates the MEN and exit from mitosis.

## Discussion

In budding yeast, the site of cytokinesis is determined long before cells undergo mitosis. Division by budding also means that the connection between mother cell and bud is small and the nucleus and other organelles must be squeezed through the bud neck to be accurately partitioned. Therefore division by budding not only requires sophisticated mechanisms to position the nucleus along the mother – bud axis, it also requires mechanisms to prevent cells from exiting mitosis and undergoing cytokinesis until the nuclei are partitioned between the mother and bud cell compartments. In 1995, Yeh et al. demonstrated the existence of such a mechanism. They showed that cells with mispositioned spindles arrest in late anaphase and fail to exit from mitosis. Subsequently, *Muhua et al. (1998)* termed this regulatory mechanism the spindle position checkpoint (SPoC). Ensuing studies showed that exit from mitosis is prevented in cells with misaligned spindles through the inhibition of the Mitotic Exit Network, the GTPase signaling cascade that promotes anaphase spindle disassembly, chromosome decondensation and cytokinesis by activating Cdc14 (*D'Aquino et al., 2005*; *Pereira and Schiebel, 2005*).

Two models have been proposed to explain how MEN activity is inhibited in response to spindle misposition: the cMT - budneck model and the zone model. The former posits that a MEN inhibitory activity is generated by a misaligned spindle, the latter that a MEN activating activity is produced by a correctly aligned spindle (*Adames et al., 2001*; *Bardin et al., 2000*; *Chan and Amon, 2010*; *Moore et al., 2009*). In this paper we took advantage of a new inducible system to study mispositioned spindles to distinguish between these two models. Our data support the conclusion that a correctly aligned spindle promotes exit from mitosis. Our data, together with previous studies, further indicate that it is the movement of the MEN component-carrying SPB into the bud that signals exit from mitosis.

## cMT – bud neck interactions do not prevent mitotic exit

Cytoplasmic microtubules continuously interact with the bud neck during spindle positioning prior to anaphase. However, once the nucleus traverses the bud neck during anaphase these interactions are lost. In cells that misposition their spindles and undergo anaphase in the mother cell, cMT – bud neck interactions persist. The proposal that it is these cMT – bud neck interactions that prevent exit from mitosis in cells with misaligned spindles stems from the analysis of cells that exit mitosis despite harboring a misaligned spindle (*Adames et al., 2001*; *D'Aquino et al., 2005*; *Pereira and Schiebel, 2005*). Previous work by Adames and Cooper demonstrated that cMT retraction from the bud neck precedes mitotic spindle breakdown in cells that exit mitosis with a mispositioned spindle (*Adames et al., 2001*). This correlation led them to propose that cMT – bud neck interactions emit an inhibitory signal that prevents exit from mitosis. This hypothesis was supported by the observations that 1) elimination of cMT – bud neck interactions by microtubule ablation or 2) loss of cMTs brought about by the cold sensitive β-tubulin allele (*tub2-401*) increased the frequency with which

cells with mispositioned spindles inappropriately exit from mitosis (*Adames et al., 2001*; *Moore et al., 2009*).

We also observed this striking correlation between cMT retraction from the bud neck and mitotic spindle breakdown, but several additional analyses demonstrate that this correlation does not indicate causality. First, retraction of cMTs occurs frequently and often for extended periods of time also in cells with-mispositioned spindles that do not inappropriately exit mitosis in the mother cell compartment. Second, in our cMT ablation studies we did not observe exit from mitosis following the loss of cMT – bud neck interactions. We cannot explain why our ablation results differ from those of *Moore et al. (2009)* but we note that mitotic exit that followed ablation of cMTs took a long time (approximately 18 min) in this previous study and regrowth of cMTs into the bud neck was also observed during the time it took until cells exited mitosis. Third, not all mutants that lack cytoplasmic microtubules exhibit an increased frequency in inappropriate exit from mitosis when the spindle is mispositioned. Gryaznova et al. (accompanying paper) found that cells lacking *SPC72* arrest in anaphase when their spindles are mispositioned despite the absence of cMTs.

The most conclusive demonstration that loss of cMT bud neck interactions does not trigger exit from mitosis was the analysis of Cdc14 localization. It clearly showed that cMT retraction from the bud neck did not precede Cdc14 release from the nucleolus but was a consequence thereof. We propose that the reason why cMTs invariably disassemble prior to the mitotic spindle in such cells is that disassembly of a single cMT upon mitotic CDK inactivation takes less time than the disassembly of a mitotic spindle that is composed of many microtubules. An inherently higher instability of cMTs compared to spindle microtubules could of course also explain this difference in disassembly timing. Together, our studies disfavor a mitotic exit inhibitory function of cMT – bud neck interactions.

## MEN activating and inhibitory zones couple exit from mitosis to spindle position

MEN signaling takes place at SPBs (*Maekawa et al., 2007*; *Valerio-Santiago and Monje-Casas, 2011*; *Visintin and Amon, 2001*). The GTPase Tem1 and Polo kinase recruit the MEN kinase Cdc15 to SPBs where it is activated by an unknown mechanism (*Rock and Amon, 2011*). Regulators of the GTPase are strategically placed in the cell. Kin4, the GTPase inhibitor localizes to the mother cell, the Kin4 inhibitor and hence MEN activator Lte1 localizes to the bud (*Bardin et al., 2000*; *D'Aquino et al., 2005*; *Pereira and Schiebel, 2005*). These localization patterns led us to propose that spindle position controls exit from mitosis through the establishment of a MEN activating compartment in the bud, a MEN inhibitory compartment in the mother cell and a sensor, the MEN component bearing SPB that shuttles between them. When both of the spindle pole bodies are in the MEN inhibitory mother cell compartment, the cell cannot exit from mitosis and arrests in anaphase. It is only once one MEN component-bearing spindle pole body moves into the mitotic exit-activating zone in the bud does the MEN become active and exit from mitosis occurs. It is important to emphasize that the zone model takes into account that not only are there MEN inhibiting factors in the mother cell compartment, but that there are also factors in the bud that promote exit from mitosis. Evidence that cells have both a negative zone in the mother cell compartment and a positive zone in the bud comes from the analysis of cells in which the MEN activating and inhibitory zones were switched. When the Kin4 inhibitor Lte1 is targeted to the mother cell, cells with mispositioned spindles inappropriately exit from mitosis (*Bardin et al., 2000*; *Bertazzi et al., 2011*). When Kin4 is targeted to the bud and its inhibitor Lte1 is inactivated, cells with correctly positioned spindles cannot exit from mitosis and arrest in anaphase (*Chan and Amon, 2010*; *Falk et al., 2011*). The analysis of heterokaryons presented here also supports the zone model. Irrespective of whether or not a cell harbors a mispositioned spindle, exit from mitosis occurs once one anaphase spindle enters the bud. Our analysis of heterokaryons in which one of the two spindles gets pulled into the bud in its entirety further shows that it is the presence of a spindle pole in bud and not a correctly positioned spindle that leads to MEN activation. In cells that harbor one nucleus in the bud and one in the mother cell, both spindles are mispositioned, yet exit from mitosis occurs once the spindle in the bud has initiated anaphase. This finding further demonstrates that two signals are necessary for MEN activation in anaphase: (1) a spatial signal – the movement of a MEN bearing SPB into the bud and (2) a temporal signal that indicates that anaphase chromosome segregation has occurred (*Bardin et al., 2000*; *Chan and Amon, 2010*; *Manzoni et al., 2010*; *Rock and Amon, 2011*). Dissecting the

molecular details of how these two signals interact to control Tem1 activity will be a critical next step in understanding how the MEN integrates spatial and temporal cues to control exit from mitosis.

## Why is the anaphase arrest caused by a mispositioned spindle not absolute?

It has long been known that a small fraction of cells exit from mitosis despite the presence of a mispositioned spindle (*Adames et al., 2001*; *D'Aquino et al., 2005*; *Pereira and Schiebel, 2005*). We show here that this event is preceded by the release of Cdc14 from the nucleolus. Our data further indicate that this Cdc14 activation requires FEAR network function because inappropriate mitotic exit in cells with mispositioned spindles is completely prevented when FEAR network component encoding genes are deleted. FEAR network activity is under the control of the regulatory mechanisms governing the metaphase – anaphase transition (*Stegmeier et al., 2002*). At this cell cycle transition, a checkpoint known as the spindle assembly checkpoint (SAC) inhibits entry into anaphase until all chromosomes have attached correctly to the mitotic spindle (reviewed in *Musacchio and Salmon (2007)*). Once this has occurred, the SAC inhibition of a protease known as Separase is alleviated and the protease initiates chromosome segregation by cleaving cohesins, the protein complexes that hold sister chromatids together (*Nasmyth, 2002*). As Separase is also a component of the FEAR network (*Stegmeier et al., 2002*), SAC activity also governs the release of Cdc14 from the nucleolus during early anaphase.

It will be interesting to determine why there are cell-to-cell differences in Cdc14 release from the nucleolus when spindles are mispositioned. Metaphase duration could be a factor. Difficulties in mitotic spindle formation and correctly attaching chromosomes to the mitotic spindle, may lead to prolonged metaphase delays, during which FEAR network component levels could increase leading to a burst of FEAR network activity once the checkpoint is satisfied and cells enter anaphase. This could also explain why the cell cycle arrest following spindle misposition is especially leaky in the *tub2-401* mutant, in which the SAC is activated. It is also possible that mitotic CDK activity, which inhibits FEAR network-mediated Cdc14 release from the nucleolus varies between individual cells. Especially high levels of activity could cause a more sustained FEAR-network-dependent release of Cdc14 from the nucleolus causing inappropriate exit from mitosis in some cells with mispositioned spindles.

Irrespective of where this variability originates from, the fact that the cell cycle arrest is not absolute in cells with mispositioned spindles is interesting. One interpretation of this observation is that FEAR network activity exhibits cell-to-cell variability with biologically meaningful consequences. It is also possible that defects in the spindle positioning pathways also subtly affect spindle position control of the MEN in some cells but not others. However, we consider this latter possibility less likely because such a scenario predicts that the few cells with mispositioned spindles that exit mitosis inappropriately do so only after a very long arrest. This is not the case.

The observation that spindle position control of the MEN and hence mitotic exit is not complete also raises the question of whether the leakiness of the arrest serves a biological function. Is it possible that under conditions where spindle misposition occurs at higher frequency (i.e. in the cold) that a not universally permanent cell cycle arrest provides an advantage? Could binucleation and hence polyploidization provide a reservoir of cells with increased adaptablity? Further investigation is needed to better understand the molecular basis and importance of these cell-to-cell differences.

## Is the spindle position checkpoint a checkpoint?

Classically, checkpoint pathways are defined as surveillance mechanisms that monitor the proceedings of a (cell cycle) event and prevent the next one from occurring until the preceding event is completed or defects therein have been corrected (*Hartwell and Weinert, 1989*). That is, an inhibitory signal prevents cells from progressing to the next cell cycle stage if the preceding cell cycle stage is still ongoing or stalled (*Rao and Johnson, 1970*). Our data show that an activating signal in the bud can override any mitotic exit inhibiting signal that may emanate from a mispositioned spindle. These data argue against a checkpoint model in the classical sense to explain the anaphase arrest in response to spindle misposition. Instead they support a model where both spatially constrained positive and negative regulatory signals control the activity of a signal transduction pathway.

Checkpoint regulation has been described for other asymmetric divisions (*Cheng et al., 2008*; *O'Connell and Wang, 2000*). In *Drosophila* male germline stem cells with mispositioned centrosomes, cell cycle progression is delayed until the centrosomes properly align with respect to the mother-daughter axis of division (*Cheng et al., 2008*). Additionally, the AMP-related kinase family member Par-1 (of which Kin4 is also a member) has been shown to be important in delaying cell cycle progression in response to mispositioned spindles in male germline stem cells (*Pereira and Yamashita, 2011*; *Yuan et al., 2012*). Given these recent findings it is tempting to speculate that a mechanism similar to the one described for spatial control of the MEN by nuclear position, rather than a checkpoint mechanism, also operates in these stem cells. The analysis of cells with multiple centrosomes analogous to what has been described here could address this question.

## Materials and methods

### Strains and plasmids

Yeast Strains are derivatives of W303 (A2587) and are described in *Supplementary file 1*. GFP-Tub1 is described in *Straight et al. (1997)*. The p*CTS1-2xmCherry-SV40NLS* plasmid was a gift from Drew Endy's lab. The *YIp211-CDC3-mCherry* plasmid was a gift from the Erfei Bi's lab and is described in *Fang et al. (2010)*. *pFA6a-link-tdTomato-SpHis5* was a gift from Kurt Thorn (Addgene plasmid # 44640). Leon Chan, Thomas Eng, and Vinny Guacci constructed the *pGPD1-OsTIR1-LEU2* and p*FA6-3V5-IAA17-KanMx6* plasmids and these were received as gifts from the D. Koshland and K. Weis labs. All gene deletions and C-terminal tags were constructed by the standard PCR-based procedures (*Longtine et al., 1998*; *Sheff and Thorn, 2004*).

### Statistics

All reported statistical error calculations are standard deviations. A biological replicate refers a replicate that was performed using the same experimental conditions but distinct yeast samples.

### Fixed cell microscopy

Indirect immunofluorescence microscopy to detect Cdc14-3HA and Tub1 was performed as described in *Kilmartin and Adams (1984)* and *(Visintin et al., 1999)*. Fixed cell microscopy of GFP-Tub1 and Cdc14-tdTomato was performed by fixing cells in a 4% paraformaldehyde and 3.4% sucrose solution for 3 min. Fixed cells were washed in potassium phosphate buffer (0.1 M KPO4, pH 7.5 and 1.2 M sorbitol) and then treated with 1% Triton for 5 min. These cells were then resuspended potassium phosphate buffer and imaged. Imaging was performed on a Zeiss Axio Observer. Z1 inverted microscope (Zeiss. Thornwood, NY) with an ORCA-ER C4742-80 CCD camera (Hamamatsu Corporation. Middlesex, NJ) and an X-Cite Series 120 arc lamp (Life Sciences & Industrial Division. Ontario, Canada). Image acquisition and analysis was performed with Molecular Devices Metamorph Software (Molecular Devices. Sunnyvale, CA).

### Live cell microscopy

Growth conditions for live cell imaging are described in the figure legends with the exception of *Figures 6* and *7* (see *prm3Δ* heterokaryon protocol below).

All imaging was done at 25°C. Imaging for the return to growth experiment (*Figure 8*) was performed with a Nikon Ti-E inverted microscope (Nikon Instruments Inc. Melville, NY) equipped with a 60X Plan APO 1.4NA objective, a GFP filter, and a CoolSNAPHQ2 CCD camera (Photometrics, Tucson, AZ), controlled by Nikon Elements software. Z stacks of four to eight sections were acquired in 10 min intervals for 12 hrs with a 12.5% ND filter and exposure times of 30-500ms.

Imaging described in *Figure 5B* was performed using Nunc Lab-Tek II Chambered Coverglass incubation chambers (Thermo Fisher Scientific. Cambridge, Mass), on a DeltaVision Elite microscope platform (GE Healthcare Bio-Sciences, Pittsburgh, PA). This microscope platform consisted of an InsightSSI solid state light source, an UltimateFocus hardware autofocus system and a model IX-71, Olympus microscope controlled by SoftWoRx software. Time-lapse images were acquired with a 60X Plan APO 1.42NA objective and a CoolSNAP HQ2 camera.

All other live cell imaging experiments, with the exception of the microtubule ablation experiment (see below), were performed on a Zeiss Axio Observer.Z1 inverted microscope (Zeiss.

Thornwood, NY) with a Heliophor Pumped Phosphor Light Engine (Chroma Technology Corp (89 North). Bellows Falls, VT). Imaging data were collected using a Hamamatsu ORCA-ER C4742-80 CCD camera (Hamamatsu Corporation. Middlesex, NJ) run by Molecular Devices Metamorph Software (Molecular Devices. Sunnyvale, CA). Cells were imaged in a CellASIC Y04C or Y04D flow cell chambers (EMD Millipore Billerica, Massachusetts).

## Cytoplasmic microtubule severing

Microtubules were cut using a Coherent OBIS 405LX laser (Coherent Inc. Santa Clara, CA). Two pre-ablation images were acquired to confirm that a cytoplasmic microtubule was present in the bud neck. The microtubule was cut with one 250 ms, 405 nm laser pulse and severing was confirmed by acquiring a Z-series of nine images spaced at 0.6 μm. Microtubule contact with the bud neck was followed at 1 min intervals for 8 min post ablation. Images of the ablated cells were acquired for up to one hour at 15 ± 1 min time intervals to determine cell cycle stage. Imaging was performed on a Nikon Eclipse Ti microscope with a Clara CCD camera (Andor Technology. South Windsor, Connecticut) and a Nikon Intensilight arc lamp.

## Generation of heterokaryons

See figure legend for culture methods to generate the *swe1Δ* heterokaryons. To generate the *prm3Δ* heterokaryons, both MATa and MATα cells were grown overnight to mid-log phase in YEPD medium at room temperature. Cells were centrifuged and resuspended in YEP and then briefly sonicated using a Branson 250 Sonifier (Branson Ultrasonics Corporation, Dansbury CT). These cells were then loaded into a Beckman elutriation rotor JE 5.0 (Beckman Coulter, Brea, CA) which was cooled to 4°C and equilibrated with YEP at 2400 rpm. Cells were loaded into the elutriation chamber at a speed to 20 mL/min and then equilibrated in the elutriator for 20–30 min at a pump speed of ∼10 mL/min. G1 cells were collected at a pump speed of ∼23 mL/min. Harvested G1 cells were concentrated using a Konte filtration system and then resuspended to a final $OD_{600nm}$ of 5.0 in YEPD. 200 μL of these cells were then plated on a YEPD agar plate and incubated at 30°C for ∼2 hr. The resultant population that contained zygotes was washed off of the agar plate using YEPD medium and loaded into a CellASIC Y04D flow chamber for time-lapse analysis.

## Acknowledgements

We would like to thank Gislene Pereira for sharing results prior to publication. We are grateful to F Solomon, Leon Chan, Anupama Seshan and members of the Amon lab for their critical reading of the manuscript. We would like to thank Rosella Visintin, Erfei Bi, Leon Chan, Karsten Weis, Doug Koshland, Thomas Eng and Vincent Guacci for strains and plasmids and Wendy Salmon for technical advice on microscopy. This work was supported in part by the Koch Institute Support (core) Grant P30-CA14051 from the National Cancer Institute. We thank the Koch Institute Swanson Biotechnology Center for reagents. This work was supported by an NIH grant (HD085866) to AA, an NIH grant (GM105755) to SL and an NIH grant to KB (R37 GM32238). AA is an investigator of the Howard Hughes Medical Institute.

## Additional information

### Funding

| Funder | Grant reference number | Author |
| --- | --- | --- |
| National Institutes of Health | HD085866 | Angelika Amon |
| Howard Hughes Medical Institute | | Angelika Amon |
| National Institutes of Health | GM105755 | Soni Lacefield |
| National Institutes of Health | R37 GM32238 | Kerry Bloom |

The funders had no role in study design, data collection and interpretation, or the decision to submit the work for publication.

## Author contributions

JEF, DT, SL, Conception and design, Acquisition of data, Analysis and interpretation of data, Drafting or revising the article, Contributed unpublished essential data or reagents; JV, Conception and design, Acquisition of data, Analysis and interpretation of data, Drafting or revising the article; KB, AA, Conception and design, Analysis and interpretation of data, Drafting or revising the article

## Author ORCIDs

Angelika Amon, iD http://orcid.org/0000-0001-9837-0314

## Additional files

### Supplementary files

• Supplementary file 1. Yeast Strain Table. The strains listed in *Supplementary file 1* are all derivatives of W303. The table shows the relevant genotype for each yeast strain and the associated strain number.

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
