## [Decision Letter]

Thank you for submitting your work entitled "Spatial signals link exit from mitosis to spindle position" for consideration by *eLife*. Your article has been favorably evaluated by Richard Losick (Senior editor) and three reviewers, one of whom, Yukiko Yamashita, is a member of our Board of Reviewing Editors.

The reviewers have discussed the reviews with one another and the Reviewing Editor has drafted this decision to help you prepare a revised submission.

Summary of comments:

This is an informative and interesting study that addresses the mechanism by which mitotic exit is controlled by the spindle orientation checkpoint in budding yeast. The study interrogates two models that have been proposed to account for the mechanism by which the successful orientation of the spindle along the mother daughter axis to support passage of one nucleus into the daughter cell triggers mitotic exit. In the first model, the association of microtubules with the bud neck region restrains mitotic exit arose. In the second, it is the physical presence of an SPB/nucleus in the bud that triggers exit. The first model was suggested following the observation of an apparent correlation between the time at which cytoplasmic microtubules lost their association with the neck region and the dissolution of the spindle (taken as a marker of mitotic exit). These former studies had been based on observations of rare events within a large population of Kar9 and Dyn1 deficient cells. In the current study the authors combine the use of the Kanemaki auxin degron system to greatly enrich for this rare category of cells within the population with more appropriate markers of mitotic exit; Cdc14 exit from the nucleolus and the departure of the septin Cdc3 from the bud neck region. The comprehensive, focused, data set that is presented reveals no correlation between the timing of the loss of microtubule association with the bud neck and mitotic exit. Rather, ingenious heterokaryon and return to cycle from meiosis experiments support the zonal model in which it is the presence of one nucleus in the bud that promotes mitotic exit. The manuscript also addresses the question of the "leakiness" of the SPOC arrest. As the abolition of the FEAR network abolishes this leakiness, the authors propose that high levels of FEAR activity enable some cells in the population to escape SPOC arrest to propagate as heterokaryons.

The reviewers agreed that the study provides a clear cut answer to two conflicting models that were proposed in the past, and that it is conducted to a high standard and support the conclusions drawn.

The reviewers raised several minor concerns that should be taken care of prior to publication as following.

1) The statement in the fourth paragraph of the Introduction that components of the FEAR network are not essential is not technically correct, as the FEAR includes Esp1 and Cdc5.

2) The Cdc14-tdTomato construct used for Figure 4 is a reason of concern. The authors state that it is slightly hypermorphic without showing the data. Yet, they claim that it accurately reflects the cell cycle changes in Cdc14 localization. Both claims should be documented by data, which could be provided as a supplemental figure. As a matter of fact, a large fraction of Cdc14-tdTomato seems to remain in the nucleolus throughout the cell cycle, which is not what the lab has shown before. Furthermore, in the Lu and Cross paper where Cdc14 release from the nucleolus was quantitatively measured using the coefficient of variation, fluorescence intensities were normalized over Net1 signal, which is constant in the nucleolus and was not used here as reference.

3) The conclusion that "cMT retraction from the bud neck is a hallmark of cells with misaligned spindles that inappropriately exit mitosis" contradicts the argument that "the majority of cells with mispositioned spindles lacked cMT-bud neck contacts for significant periods of time yet stayed arrested in anaphase".

4) In Figure 1, in the left cell with a mispositioned spindle, Tem1 is depicted as two yellow dots that seem to float in the cytoplasm. Work from other groups, including one of the present authors, has shown that in cells with mispositioned spindles Tem1 is symmetrically localized at SPBs (Pereira et al., 2000; Molk et al., 2004; Pereira and Caydasi, 2009). Thus, the cartoon should be modified accordingly.

5) In Figure 6, right cell, cMTs should be shown to be interacting with the bud neck, since this is the signal that is supposed to inhibit mitotic exit according to the alternative model.

6) In the second paragraph of the subsection “A system to monitor spindle misposition by live cell microscopy”, it is mentioned that the existence of minor spindle positioning pathways could explain the ability of *DYN1-AID kar9Δ* cells to eventually re-position the spindle. Indeed, an auxiliary spindle positioning process independent of Kar9 and dynein has been recently described (Kirchenbauer and Liakopoulos, 2013, MBoC 24: 1434) and could be cited here.

7) The requirement of the FEAR pathway for the unscheduled mitotic exit of cells in the presence of mispositioned spindles has been published last year by another group (Scarfone et al., 2015, PloS Gen. 11: e1004938). This work should be quoted here.

8) The authors focus on a minor population of cells: those that exit mitosis despite having mispositioned spindle. They state that why this happens remain unclear. Although the data provided in the manuscript is convincing on their own, I have a concern (though minor) that the interpretation relies on low frequency events for which cause remains unknown. Why do those cells exit mitosis inappropriately? Are *dyn1/kar9* mutant cells also defective in SPOC to some extent (for whatever unknown reasons)? I understand that *dyn1/kar9* that eventually exit mitosis with correct spindle position don't provide insights into the question being asked in this manuscript, but I think a caution is required to use low frequency events (of unknown cause) to build logic, and I'd suggest adding an explanation about this in the manuscript.

---

## [Author Response]

The reviewers raised several minor concerns that should be taken care of prior to publication as following.

*1) The statement in the fourth paragraph of the Introduction that components of the FEAR network are not essential is not technically correct, as the FEAR includes Esp1 and Cdc5.*

Agreed. We have revised the wording accordingly.

*2) The Cdc14-tdTomato construct used for Figure 4 is a reason of concern. The authors state that it is slightly hypermorphic without showing the data. Yet, they claim that it accurately reflects the cell cycle changes in Cdc14 localization. Both claims should be documented by data, which could be provided as a supplemental figure. As a matter of fact, a large fraction of Cdc14-tdTomato seems to remain in the nucleolus throughout the cell cycle, which is not what the lab has shown before. Furthermore, in the Lu and Cross paper where Cdc14 release from the nucleolus was quantitatively measured using the coefficient of variation, fluorescence intensities were normalized over Net1 signal, which is constant in the nucleolus and was not used here as reference.*

We now provide an in depth characterization of the Cdc14-tdTomato construct. We determined the fraction of cells with mispositioned spindles that exit mitosis when harboring this construct. This analysis, presented in Figure 4—figure supplement 1, shows that approximately 25% of Cdc14-tdTomato cells with mispositioned spindles exit mitosis inappropriately, indicating that the allele is hypermorphic.

To carefully compare the kinetics of Cdc14-tdTomato release from the nucleolus with that of the non-hypermorphic Cdc14-3HA allele that we used in previous studies, we analyzed the timing of Cdc14 release of from the nucleolus as a function of spindle length. This analysis, presented in Figure 4—figure supplement 1, shows that both fusion proteins are released from the nucleolus with almost identical kinetics during anaphase.

We also provide additional data to demonstrate that the release of Cdc14-tdTomato from the nucleolus in cells with mispositioned spindles is not simply due to changes in nucleolar morphology. As suggested, we compared the localization of a nucleolar marker (we used Cfi1-GFP) with the localization of Cdc14-tdTomato in cells also harboring a tubulin-GFP fusion. As shown in the montage in Figure 4—figure supplement 3 and Figure 4—figure supplement 4, Cdc14 nucleolar intensity decreases during anaphase before cMTs retract from the bud neck, but Cfi1 nucleolar intensity remains unchanged.

We attempted normalize the CV of the Cdc14-tdTomato intensity to that of Cfi1-GFP but this quantification was not successful because the Cfi1-GFP signal overlapped with that of the tubulin-GFP signal. We next attempted to perform this normalization using a Cfi1-BFP but this fusion was not detectable in live-cell imaging. Lastly, we tried to quantify the Cfi1-GFP signal in cells were only the (+) ends of microtubules were marked with a Bim1-GFP fusion. However, in this set up we were not able to reliably identify cytoplasmic microtubules because the Bim1-GFP signal was too dim. This prevented us from determining when cMTs retract from the bud neck. Because of these complications we are only able to show a qualitative comparison of the Cdc14-tdTomato and Cfi1-GFP signal. The reasons of why we were not able to perform this normalization are also mentioned in the text.

*3) The conclusion that "cMT retraction from the bud neck is a hallmark of cells with misaligned spindles that inappropriately exit mitosis" contradicts the argument that "the majority of cells with mispositioned spindles lacked cMT-bud neck contacts for significant periods of time yet stayed arrested in anaphase".*

Fixed.

*4) In Figure 1, in the left cell with a mispositioned spindle, Tem1 is depicted as two yellow dots that seem to float in the cytoplasm. Work from other groups, including one of the present authors, has shown that in cells with mispositioned spindles Tem1 is symmetrically localized at SPBs (Pereira et al., 2000; Molk et al., 2004; Pereira and Caydasi, 2009). Thus, the cartoon should be modified accordingly.*

We previously showed that Tem1-13MYC, a non-hypermorphic tagged *TEM1* allele, is very rarely detected on SPBs in cells with mispositioned spindles (D'Aquino, et al., 2005). It is our experience that GFP fusions tend to have a higher affinity to SPBs and so caution is warranted when analyzing these constructs. We further found that a Tem1-eGFP fusion is hypermorphic (Chan and Amon, 2009). Together, these observations lead us to believe that the localization of the Tem1-13MYC fusion likely reflects the true localization of Tem1 in cells with mispositioned spindles, that is off the SPBs. We have explained this in the figure legend but kept the figure as is.

*5) In Figure 6, right cell, cMTs should be shown to be interacting with the bud neck, since this is the signal that is supposed to inhibit mitotic exit according to the alternative model.*

To eliminate redundancies in the paper we removed Figure 6.

*6) In the second paragraph of the subsection “A system to monitor spindle misposition by live cell microscopy”, it is mentioned that the existence of minor spindle positioning pathways could explain the ability of DYN1-AID kar9Δ cells to eventually re-position the spindle. Indeed, an auxiliary spindle positioning process independent of Kar9 and dynein has been recently described (Kirchenbauer and Liakopoulos, 2013, MBoC 24: 1434) and could be cited here.*

Thanks for pointing this out and done.

*7) The requirement of the FEAR pathway for the unscheduled mitotic exit of cells in the presence of mispositioned spindles has been published last year by another group (Scarfone et al., 2015, PloS Gen. 11: e1004938). This work should be quoted here.*

Done.

8) The authors focus on a minor population of cells: those that exit mitosis despite having mispositioned spindle. They state that why this happens remain unclear. Although the data provided in the manuscript is convincing on their own, I have a concern (though minor) that the interpretation relies on low frequency events for which cause remains unknown. Why do those cells exit mitosis inappropriately? Are dyn1/kar9 mutant cells also defective in SPOC to some extent (for whatever unknown reasons)? I understand that dyn1/kar9 that eventually exit mitosis with correct spindle position don't provide insights into the question being asked in this manuscript, but I think a caution is required to use low frequency events (of unknown cause) to build logic, and I'd suggest adding an explanation about this in the manuscript.

This is a very interesting possibility that we should have considered in the Discussion. The reviewer is correct that we cannot exclude the possibility that the two spindle positioning systems play a minor role in preventing exit from mitosis in the event of spindle misposition. We have added text considering this possibility to the Discussion. However, we should add that if this were the case one would predict that the few cells with mispositioned spindles that exit mitosis inappropriately do so only after a very long arrest. This is however not always true, which would argue against this possibility.